# A protein interaction mechanism for suppressing the mechanosensitive Piezo channels

Tingxin Zhang[1,2], Shaopeng Chi[1], Fan Jiang[1], Qiancheng Zhao[1] & Bailong Xiao[1]

Piezo proteins are bona fide mammalian mechanotransduction channels for various cell types including endothelial cells. The mouse Piezo1 of 2547 residues forms a three-bladed, propeller-like homo-trimer comprising a central pore-module and three propeller-structures that might serve as mechanotransduction-modules. However, the mechanogating and regulation of Piezo channels remain unclear. Here we identify the sarcoplasmic /endoplasmic-reticulum $Ca^{2+}$ ATPase (SERCA), including the widely expressed SERCA2, as Piezo interacting proteins. SERCA2 strategically suppresses Piezo1 via acting on a 14-residue-constituted intracellular linker connecting the pore-module and mechanotransduction-module. Mutating the linker impairs mechanogating and SERCA2-mediated modulation of Piezo1. Furthermore, the synthetic linker-peptide disrupts the modulatory effects of SERCA2, demonstrating the key role of the linker in mechanogating and regulation. Importantly, the SERCA2-mediated regulation affects Piezo1-dependent migration of endothelial cells. Collectively, we identify SERCA-mediated regulation of Piezos and the functional significance of the linker, providing important insights into the mechanogating and regulation mechanisms of Piezo channels.

[1] School of Pharmaceutical Sciences or Life Sciences, Tsinghua-Peking Joint Center for Life Sciences, IDG/McGovern Institute for Brain Research, Tsinghua University, Beijing 100084, China. [2] Joint Graduate Program of Peking-Tsinghua-NIBS, School of Life Sciences, Tsinghua University, Beijing 100084, China. Tingxin Zhang and Shaopeng Chi contributed equally to this work. Correspondence and requests for materials should be addressed to B.X. (email: xbailong@biomed.tsinghua.edu.cn)

Mechanosensitive (MS) ion channels are specialized mechanotransducers that rapidly convert mechanical force into electrochemical signals, a process known as mechanotransduction[1–3]. The identification of the evolutionarily conserved Piezo proteins, including Piezo1 and Piezo2, as the bona fide mammalian MS cation channels[4,5], has heralded a new era of research on mechanotransduction mechanisms in mammals[6,7]. On the one hand, Piezo proteins have been demonstrated to serve as physiologically and pathophysiologically important mechanotransducers for a wide variety of mechanotransduction processes[1]. For instance, Piezo1 has been proposed to function as the blood flow sensor required for endothelium-dependent vascular development and blood pressure regulation[8–10]. Additionally, it plays critical roles in red blood cell volume regulation[11,12], cell migration[13], homeostasis of epithelial cell numbers[14,15], high-strain mechanotransduction of cartilage[16,17], regulation of urinary osmolarity[18], neural stem cell fate determination[19], and neuronal axon guidance[20]. Piezo2 functions as a major mechanotransduction channel for sensation of gentle touch[21–24], proprioception[25], airway stretch and lung inflation[26]. Furthermore, mutations of human Piezo1 or Piezo2 gene have been linked to a number of genetic diseases[1], demonstrating the importance of Piezo-associated mechanotransduction in pathophysiology. On the other hand, Piezo1, a complicated transmembrane protein without sequence homology to other known ion channels, has been shown to form a unique three-bladed, propeller-shaped homo-trimeric channel complex of ~ 0.9 Mega-dalton, comprising a central ion-conducting pore and three peripheral propeller-resembling structures[27,28]. The peripheral structures might function as mechanotransduction modules to confer mechanosensitivity to the central pore[27,28]. Nevertheless, it remains unclear how the two distinct modules are effectively coupled and how Piezo channels are regulated.

Here we find that the family of sarcoplasmic /endoplasmic-reticulum Ca$^{2+}$ ATPase (SERCA), including SERCA1-3, interacts with Piezo proteins. Focusing on characterizing the interaction and regulation between Piezo1 and SERCA2, we have identified that a 14-residue-constituted intracellular linker region between the pore-module and the mechanotransduction-module is critically required for mechanogating and SERCA2-mediated suppression of Piezo1. These findings suggest that the linker region might play a key role in coupling the mechanotransduction-module to the pore-module, in analogous to the role of the S4-S5 linker of the voltage-gated K$^+$ channels or the transient receptor potential (TRP) channels[29,30]. Thus, our studies support the working model that Piezo1 employs the peripheral propeller-structures as mechanotransduction modules to gate the central ion-conducting pore-module[27,28]. Furthermore, we show that the

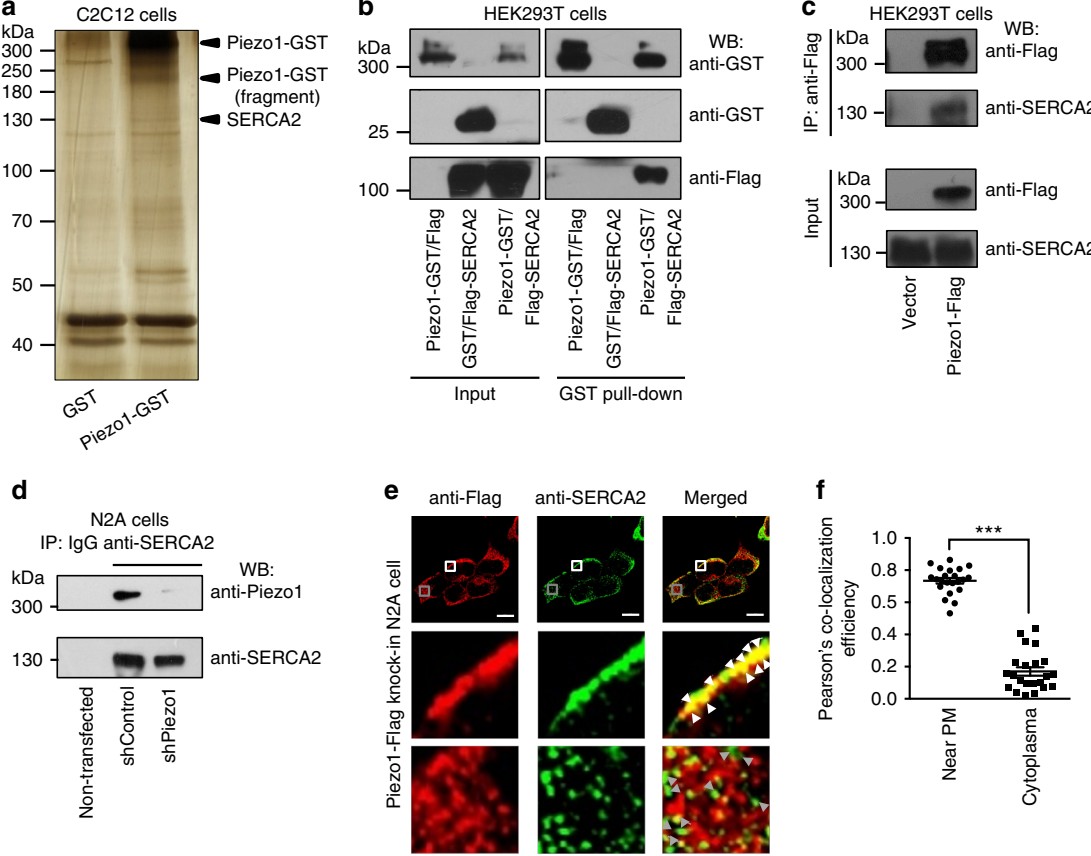

**Fig. 1** Identification of SERCA2 as an interacting protein of Piezo1 **a**, A representative silver-staining gel showing the protein bands indicated with arrow heads specifically presented in the Piezo1-GST pulled-down sample that were subjected to mass spectrometry for identification. **b**, Western blots showing the interaction of over-expressed Piezo1-GST and Flag-SERCA2 in HEK293T cells (repeated 3 times). **c**, Western blots showing pull-down of endogenous SERCA2 by overexpressed Piezo1-Flag proteins in HEK293T cells (repeated 3 times). **d**, Western blots showing co-immunoprecipitation of endogenously expressed Piezo1 and SERCA2 in N2A cells (repeated 3 times). **e**, Immunofluorescent staining images of endogenous Piezo1 and SERCA2 in the Piezo1-Flag knock-in N2A cell line (repeated 3 times). The white and grey boxes respectively illustrate the staining of Piezo1 and SERCA2 either near plasma membrane or in the cytoplasm. Scale bar, 10 μm. **f**, Pearson's co-localization efficiency for Piezo1 and SERCA2 located either near plasma membrane or in the cytoplasm that was calculated by correlation intensity analysis of the anti-Flag and anti-SERCA2 signals. Each dot represents an individual cell (22 cells in total) and short horizontal lines indicate the mean ± s.e.m. Unpaired student's t-test, ***$p < 0.001$

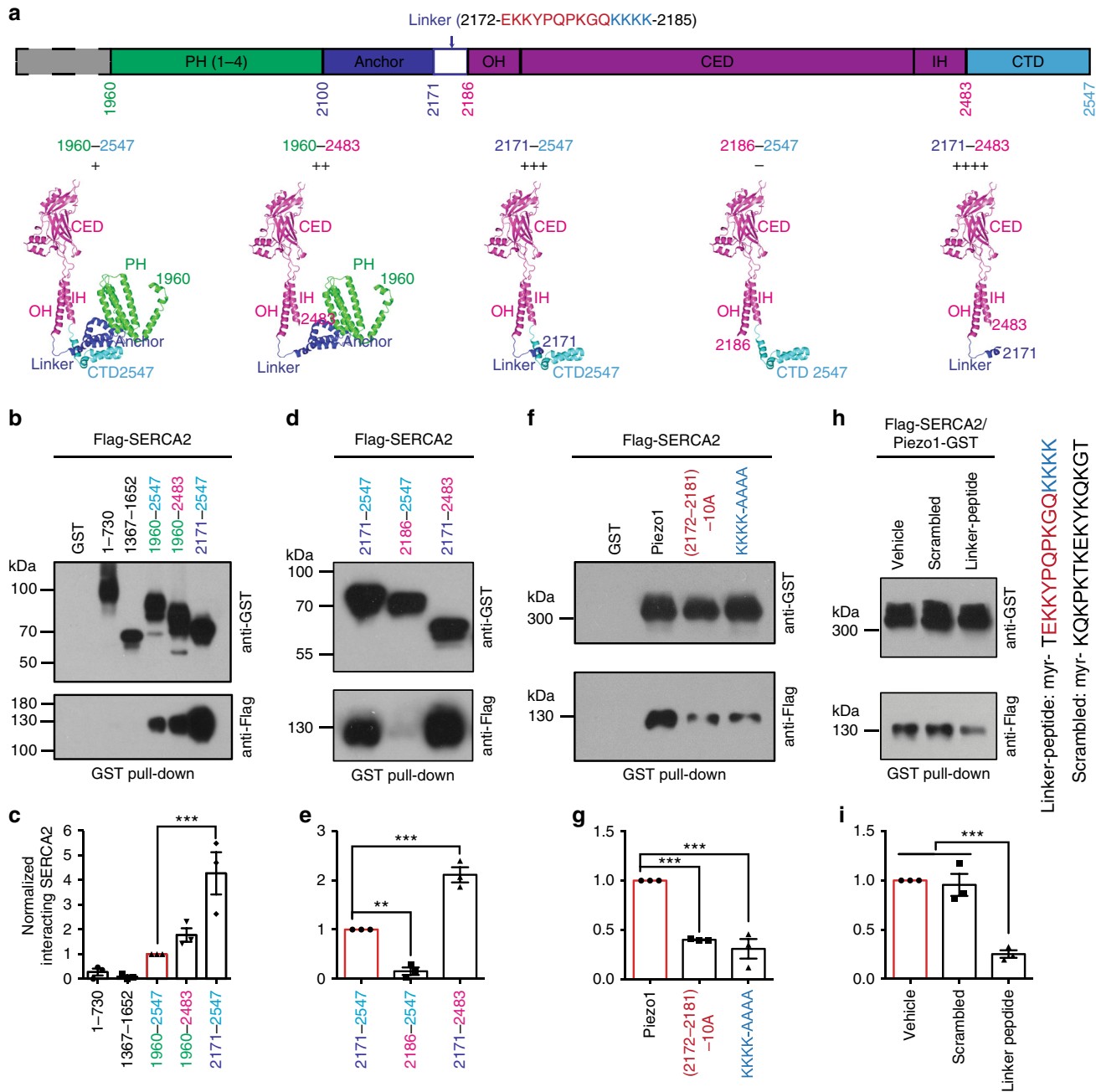

**Fig. 2** The linker region 2172–2185 of Piezo1 that connects the central pore and the peripheral propeller-structure is required for SERCA2 interaction. **a**, Topological and structural illumination of the C-terminal regions of Piezo1 (1960–2547) including the specified featured domains (adopted from PDB 4RAX for the atomic CED structure and 3JAC for the rest of the structure as alanine model[27]). The residues of the linker region 2172–2185 are shown. The symbols of + and – indicate the relative strength of the interaction. **b**, **d**, **f**, Western blots showing GST pull-down of the co-expressed Flag-SERCA2 proteins using the indicated GST-fused Piezo1 fragment proteins or mutants (repeated 3 times). **h**, Western blots showing Piezo1-GST pull-down of co-expressed Flag-SERCA2 in the presence of the indicated conditions. The amino acid composition of the linker-peptide and the scrambled peptide are shown in the right (repeated 3 times). **c**, **e**, **g**, **i**, Scatter plot of the interacting SERCA2 level shown in **b**, **d**, **f**, **h**, respectively (normalized to the samples represented by the red bars). One-way ANOVA with multiple comparison test. Data shown as mean ± s.e.m. $^{**}p < 0.01$, $^{***}p < 0.001$

SERCA2-mediated regulation of Piezo1 affects the Piezo1-dependent migration of endothelial cells.

## Results

**Identification of SERCAs as interacting proteins of Piezo1.** To identify regulatory proteins of Piezo1, we reasoned that interacting proteins more likely exist in cells that express relatively high level of endogenous Piezo1, such as C2C12 cells. Therefore, we first carried out glutathione-S-transferase (GST) pull-down from HEK293T cells (negligible expression of endogenous Piezo1) transfected with constructs encoding either the GST control or the Piezo1-GST fusion protein. The GST or Piezo1-GST associated beads were then incubated with cell lysates derived from C2C12 cells, followed with silver staining of the pulled-down proteins after separation on a SDS-PAGE gel. We detected a strong band that matches the molecular weight of the Piezo1-GST protein (~ 300 kDa) specifically in the Piezo1-GST sample, but not in the GST control sample (Fig. 1a), indicating a

successful pull-down of the Piezo1-GST proteins. Two other protein bands near the molecular markers of 250 kDa or 130 kDa were specifically visualized in the Piezo1-GST sample group (Fig. 1a). These three bands and the corresponding bands from the GST-control sample were excised for mass spectrometry analysis. In both the 300-kDa- and 250-kDa-band samples, peptides corresponding to mouse Piezo1 were identified, indicating that these two bands contained full-length and partially degraded Piezo1 proteins, respectively. Interestingly, in the 130-kDa-band sample, we identified peptides that correspond to the isoform 2 of the sarcoplasmic/endoplasmic-reticulum $Ca^{2+}$ ATPase (SERCA2) (Supplementary Table 1 and Supplementary Data 1), which has a molecular weight of ~ 110 kDa. SERCA2 is a SR/ER-localized $Ca^{2+}$ ATPase for recycling cytosolic $Ca^{2+}$ into the SR/ER $Ca^{2+}$ store, a process essential for maintaining $Ca^{2+}$ homeostasis in nearly all cell types including endothelial cells[31].

To verify whether SERCA2 is indeed a binding protein of Piezo1, we generated the Flag-tagged SERCA2 (Flag-SERCA2) construct that encodes the widely expressed splicing variant of SERCA2, SERCA2b[31]. Pull-down experiments from HEK293T cells co-transfected with Piezo1-GST and Flag-SERCA2 revealed that the Flag-SERCA2 proteins were pulled down together with the Piezo1-GST proteins using the glutathione beads (Fig. 1b). By contrast, no Flag-SERCA2 proteins were detected from pulled-down samples derived from cells transfected with either Piezo1-GST alone or GST/Flag-SERCA2 (Fig. 1b). Similarly, SERCA2a, the other splicing variant of SERCA2 that is majorly expressed in cardiomyocytes, was also able to interact with Piezo1 (Supplementary Fig. 1). In addition, the two other isoforms of the SERCA family, SERCA1 (mainly expressed in skeletal muscles) and SERCA3 (expressed in a limited number of non-muscle cells), are also able to interact with Piezo1 (Supplementary Fig. 1). Given the wide expression of SERCA2, we next focused on characterizing the interaction between SERCA2 and Piezo1.

Consistent with the observation that the recombinant Piezo1-GST proteins were able to pull down endogenously expressed SERCA2 in C2C12 cells (Fig. 1a), we found that the heterologously over-expressed Piezo1-Flag proteins were able to pull down endogenously expressed SERCA2 proteins in HEK293T cells, which were detected via western blotting with the anti-SERCA2 antibody (Fig. 1c). We next used the anti-SERCA2 antibody to pull down endogenously expressed SERCA2 from Neuro2A (N2A) cells, which express relatively high level of Piezo1 proteins[4]. In the anti-SERCA2-immunoprecipitated sample, a protein band near the 300 kDa molecular marker was specifically detected by the Piezo1 antibody (Fig. 1d). shRNA-mediated knockdown of Piezo1 drastically reduced the level of this co-immunoprecipitated protein, verifying the specific existence of the Piezo1 proteins in the Piezo1-antibody-recognized band in the control shRNA-transfected group (Fig. 1d). These pull-down experiments demonstrate that both heterologously and endogenously expressed Piezo1 and SERCA2 can interact with each other.

Piezo1 functions as a mechanosensitive cation channel in the plasma membrane (PM), while SERCA2 is an ER $Ca^{2+}$ pump. We therefore employed immunofluorescent staining to examine the endogenous co-localization between Piezo1 and SERCA2. To stain endogenously expressed Piezo1 proteins, we used the CRISPR/Cas9 technology to generate a Piezo1-Flag knock-in N2A cell line (Supplementary Fig. 2). The Flag-encoding sequence was inserted in the C-terminal extracellular domain (CED) after the position corresponding to the residue G2420 of mouse Piezo1 (Supplementary Fig. 2a, b). The correct expression of the Piezo1-Flag protein in the knock-in cell line was verified by specific anti-Flag immunostaining (Supplementary Fig. 2c).

When assayed by single-cell Fura-2 $Ca^{2+}$ imaging and electrophysiology, the Piezo1-Flag knock-in cells showed similar Yoda1 (a Piezo1 chemical activator[32])-evoked $Ca^{2+}$ responses (Supplementary Fig. 2d, e) and Piezo1-mediated poking-induced currents (Supplementary Fig. 2f–h) as the wild-type N2A cells did, demonstrating the normal functionality of the endogenous Piezo1-Flag proteins.

Co-immunostaining of the knock-in cells with the anti-Flag and anti-SERCA2 antibodies and subsequent confocal imaging revealed high level of co-localization of Piezo1 and SERCA2 at the periphery of the cell (white box of Fig. 1e and Fig. 1f). Piezo1 proteins were also detected inside the cell, where they showed less co-localization with SERCA2 (gray box of Fig. 1e and Fig. 1f). These data suggest that Piezo1 and SERCA2 might interact at the PM-ER junction, in analogous to the interaction between the ER-localized STIM1 and PM-localized Orai proteins that constitute the $Ca^{2+}$ release activated $Ca^{2+}$ (CRAC) channel[33].

**The Piezo1 linker region is required for SERCA2 interaction**. We next set out to identify the region in Piezo1 that is responsible for interacting with SERCA2. We found that the C-terminal fragment of Piezo1 (1960–2547) is capable of pulling down the co-expressed Flag-SERCA2 protein (Fig. 2a, b). By contrast, both the N-terminal fragment (1–730) and the predicted intracellular fragment located in the central region (1367–1652) were ineffective (Fig. 2b). The fragment of 1960–2547 contains the structurally resolved peripheral helix 1–4 (PH1-4), the Anchor, the linker and the pore-module that includes the outer helix (OH), C-terminal extracellular domain (CED), inner helix (IH), and C-terminal intracellular domain (CTD) (Fig. 2a). Intriguingly, removing either the CTD (2484–2547) or the PH/Anchor (1960–2170) resulted in even more robust pull-down of SERCA2 by the corresponding fragments of 1960–2483 and 2171–2547, respectively (Fig. 2a–c), indicating that the PH/Anchor and CTD domains might render steric hindrance for SERCA2 interaction.

Based on the structural organization (Fig. 2a), the intracellularly located lysine-rich linker region (2172–2185) that connects the Anchor and OH is exposed to the intracellular surface, but is partially masked by the CTD (Fig. 2a). Therefore the linker region could serve as a binding element for SERCA2. In line with this hypothesis, the linker-containing fragments of 2171–2483 (without CTD) and 2171–2547 (with CTD) were able to interact with SERCA2, while the linker-free fragment of 2186–2547 showed nearly abolished interaction (Fig. 2a, d, e). Furthermore, the fragment of 2171–2483 without CTD appeared to have stronger interaction with SERCA2 than the fragment of 2171–2547 with CTD (Fig. 2a, d, e), in line with partially masking the linker region by the intracellular CTD.

We went on to examine whether the 14-residue-constituted linker region is required for the interaction between SERCA2 and the full-length Piezo1. Neutralizing either the residues 2172–2181 [Piezo1-(2172–2181)10A] or the cluster of 4 lysine residues (2182–2185) (Piezo1-KKKK-AAAA) in Piezo1 to alanine reduced SERCA2-Piezo1 interaction (Fig. 2f, g). These data demonstrate that the residues in the linker region are required for the interaction between Piezo1 and SERCA2.

Given that the linker region is critically required for SERCA2 interaction to both the full-length Piezo1 and the structurally defined C-terminal fragments, we hypothesized that the linker likely serves as a direct binding site for SERCA2. To validate this hypothesis, we synthesized the linker-peptide (2171–2185) and the scrambled-peptide with myristoylation at the N-terminal residue for allowing membrane penetration and then tested their effect on Piezo1-SERCA2 interaction. The linker-peptide, but not the scrambled-peptide, reduced the interaction between Piezo1

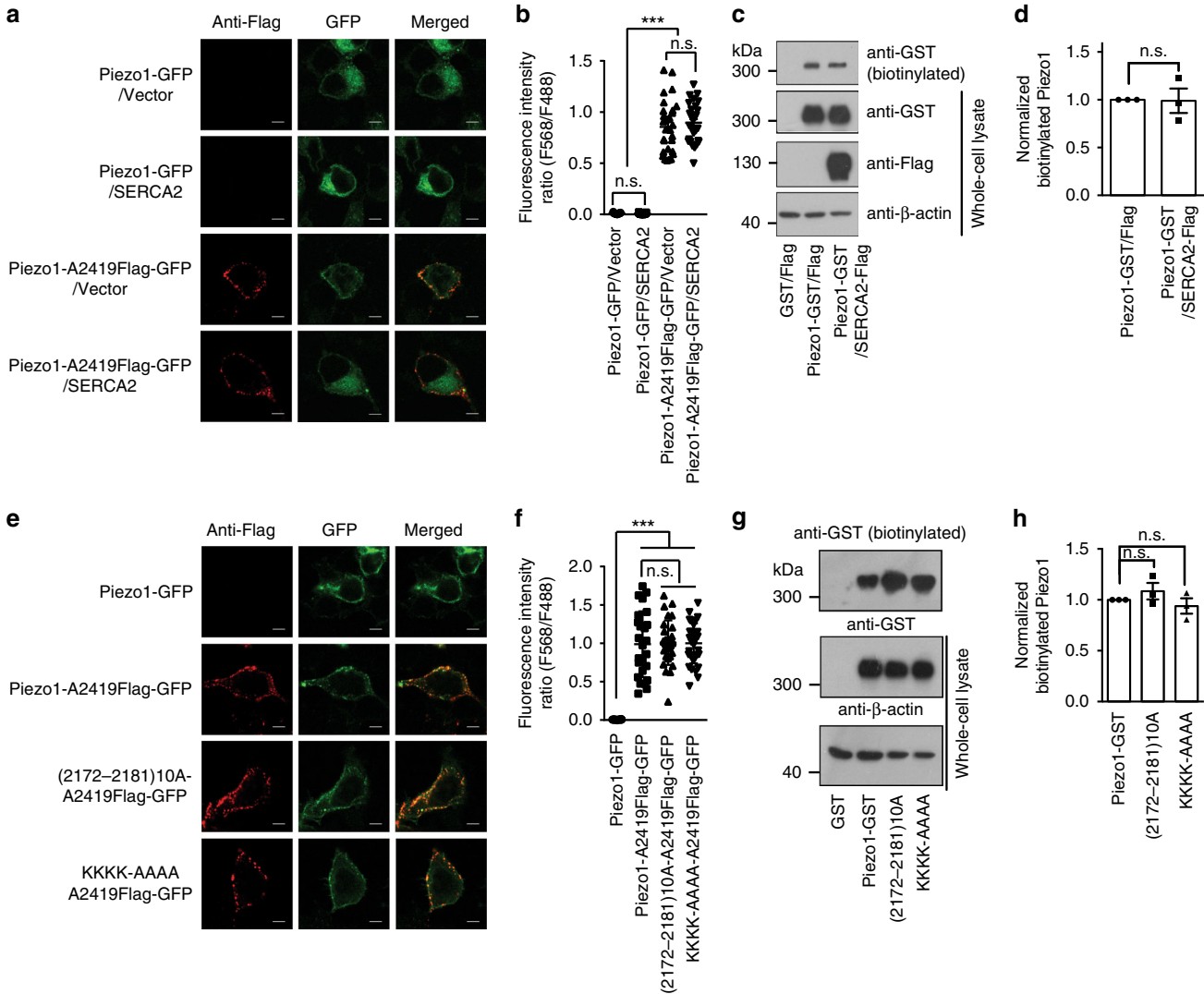

**Fig. 3** Neither SERCA2 co-expression nor the linker-mutations affect the expression of Piezo1 in plasma membrane. **a** and **e**, Live immunofluorescent staining of the extracellularly localized Flag-tag inserted after the residue A2419 of the Piezo1-GFP, 2172–2181(10A)-GFP, and KKKK/AAAA-GFP fusion proteins from HEK293T cells transfected with the indicated constructs. The GFP images were taken as control for the expression of the fusion proteins. Scale bar, 5 μm. **b** and **f**, Scatter plots of the fluorescence intensity ratio of the anti-Flag signal (F568) over GFP signal (F488). Each dot represents the ratio of F568/F488 from an individual cell. One-way ANOVA with multiple comparison test. **c** and **g**, Western blots of the biotinylated or whole-cell lysate samples derived from HEK293T cells transfected with the indicated constructs. **d** and **h**, Scatter plots of the normalized biotinylated Piezo1 levels of cells transfected with the indicated constructs. Unpaired student's t-test (**d**) or One-way ANOVA with multiple comparison test (**h**). Data shown as mean ± s.e. m. ***$p < 0.001$

and SERCA2 (Fig. 2h, i), indicating that the linker-peptide and Piezo1 compete for SERCA2 interaction. Collectively, these data suggest that the linker region serves as a critical binding site for SERCA2.

The identification of the key interacting residues in Piezo1 provides compelling evidence that SERCA2 might directly bind to Piezo1. This differs from previously identified Piezo1 regulatory proteins including polycystein-2 (PC-2) and stomatin-like protein-3 (STOML3), which appears to regulate Piezo function through indirectly altering the membrane curvature or stiffness[34–36]. We thus went on to test how SERCA2 interaction could regulate Piezo1.

**No effect of SERCA2 or the mutations on Piezo1 localization**. We first examined whether the plasma membrane expression of Piezo1 is affected by SERCA2 co-expression or mutating the linker region (Fig. 3a). We inserted a Flag tag after A2419 located

in the extracellular CED[28] into the Piezo1-GFP, 2172–2181(10A)-GFP and KKKK/AAAA-GFP fusion constructs (Piezo1-A2419Flag-GFP, 2172–2181(10A)-A2419Flag-GFP and KKKK/AAAA-A2419Flag-GFP, respectively), and then carried out live immunostaining of the Flag tag from HEK293T cells transfected with the constructs without permeabilizing the membrane. The GFP images were taken as control for the expression of the fusion proteins. As shown in Fig. 3a, e, the anti-Flag immunofluorescent signal was specifically detected at the periphery of cells transfected with the Flagged-constructs, but not from Piezo1-GFP-expressing cells. Quantitative analysis of the fluorescence intensity ratio of the anti-Flag signal over the GFP signal revealed that neither co-expression of SERCA2 nor mutating the linker region affected the plasma membrane expression of Piezo1 (Fig. 3b, f).

To validate the result, we carried out cell surface protein biotinylation assay. Western blotting of the Piezo1-GST, 2172–2181(10A)-GST and KKKK/AAAA-GST proteins in the

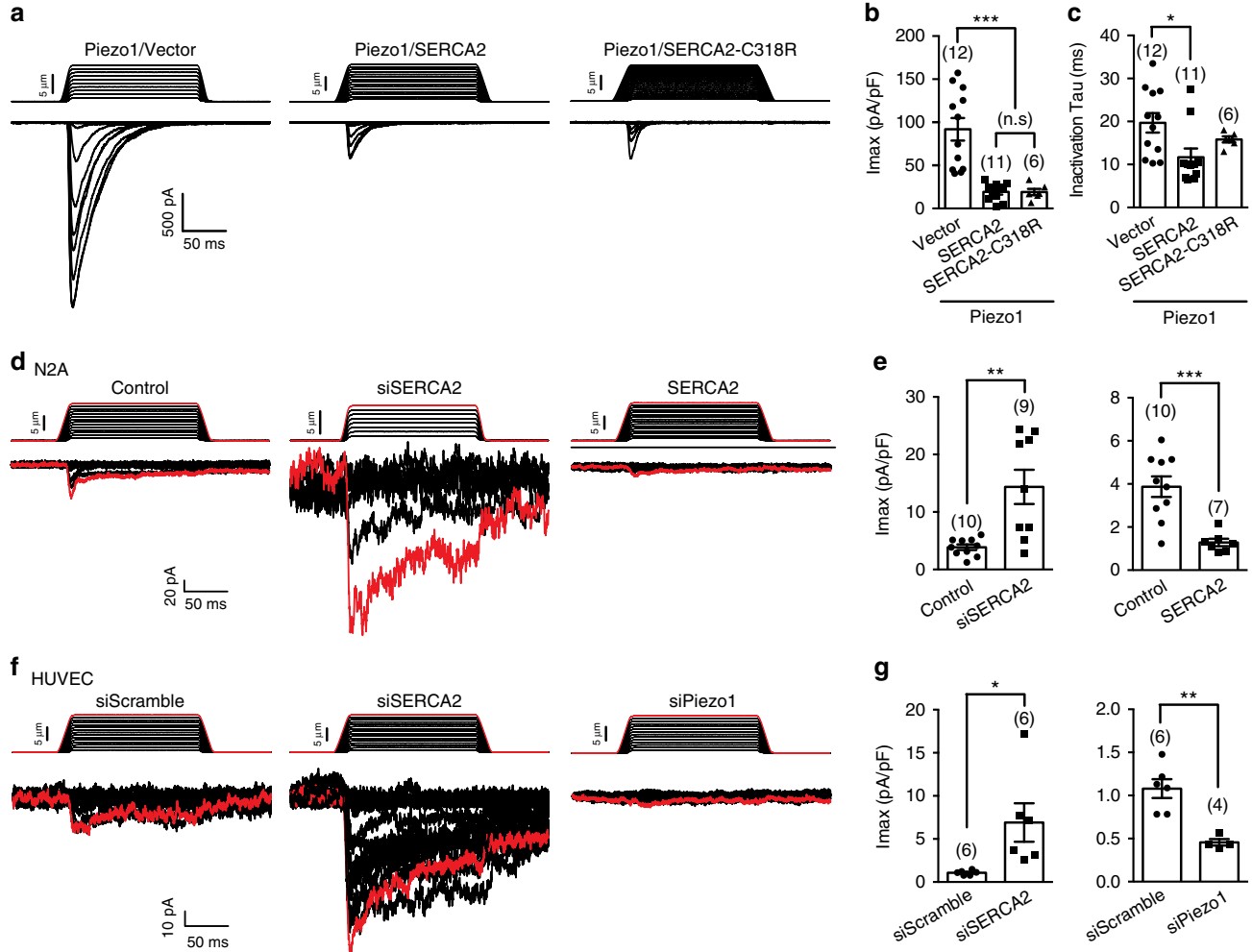

**Fig. 4** SERCA2 inhibits Piezo1-mediated poking-induced currents. **a**, Representative traces of poking-induced inward currents recorded at -60 mV in HEK293T cells with the indicated transfections. **b** and **c**, Scatter plots of the maximal poking-induced currents (**b**) and inactivation tau (**c**) of the indicated transfections. One-way ANOVA with multiple comparison test. **d** and **f**, Representative current traces of poking-induced inward currents recorded at -60 mV of either N2A (**d**) or HUVEC (**f**) cells transfected with the indicated conditions. **e** and **g**, Scatter plots of the maximal poking-induced currents of either N2A (**e**) or HUVEC (**g**) cells transfected with the indicated conditions. Unpaired student's t-test. Data shown as mean ± s.e.m. $^*p < 0.05$, $^{**}p < 0.01$, $^{***}p < 0.001$

whole-cell lysate revealed that their overall expression neither affected by co-expression of SERCA2 (Fig. 3c) nor by the linker mutations (Fig. 3g). Furthermore, western blotting of the biotinylated protein samples in plasma membrane pulled-down via streptavidin-beads shows similar level of biotinylated Piezo1-GST proteins with or without SERCA2 (Fig. 3c, d) or between wild type and the linker mutants of Piezo1 (Fig. 3g, h). These results are in line with the live immunofluorescent results (Fig. 3a, b, e, f). Collectively, these data suggest that SERCA2 interaction or mutating the linker region does not affect the plasma membrane expression of Piezo1.

**SERCA2 suppresses Piezo1-mediated mechanosensitive currents**. We next focused on characterizing the effect of SERCA2-Piezo1 interaction on Piezo1 channel function. Co-expression of SERCA2 with Piezo1 drastically suppressed the poking-induced maximal whole-cell currents (Piezo1/Vector vs Piezo1/SERCA2: 91.9 ± 13.1 vs 19.2 ± 3.1 pA/pF) and fastened the inactivation rate (Piezo1/Vector vs Piezo1/SERCA2: 19.7 ± 2.3 vs 11.7 ± 2.0 ms) (Fig. 4a–c). Furthermore, the $Ca^{2+}$-pumping-deficient mutant SERCA2-C318R[37] (Supplementary Fig. 3a, b), which had no

effect on the expression of the co-transfected Piezo1 (Supplementary Fig. 3c), remained effective in suppressing Piezo1-mediated poking-induced currents (19.1 ± 3.7 pA/pF for Piezo1/SERCA2-C318R) (Fig. 4a–c). These data suggest that the suppressive effect of SERCA2 on Piezo1 was not dependent on its $Ca^{2+}$ pumping activity.

We next examined whether endogenous Piezo1-mediated mechanosensitive currents can be regulated by SERCA2. Consistent with the previous studies with N2A cells[4], poking-induced a step-dependent inward current with a maximal current of 3.9 ± 0.5 pA/pF (Fig. 4d, e), which was significantly reduced upon Piezo1 knockdown (Supplementary Fig. 2f, g). siRNA-mediated knockdown of endogenous SERCA2 (Supplementary Fig. 3d) enhanced the current to 14.4 ± 3.0 pA/pF (Fig. 4d, e). By contrast, overexpression of SERCA2 suppressed the endogenous Piezo1 currents to 1.3 ± 0.2 pA/pF (Fig. 4d, e). These data demonstrate that endogenous Piezo1-mediated mechanosensitive currents in N2A cells are functionally regulated by SERCA2.

Piezo1 is expressed in endothelial cells for proper vascular development and blood pressure regulation[8,9,38], promoting us to investigate the regulation of Piezo1 by SERCA2 in this cell type. In human umbilical vein endothelial cells (HUVEC), we detected

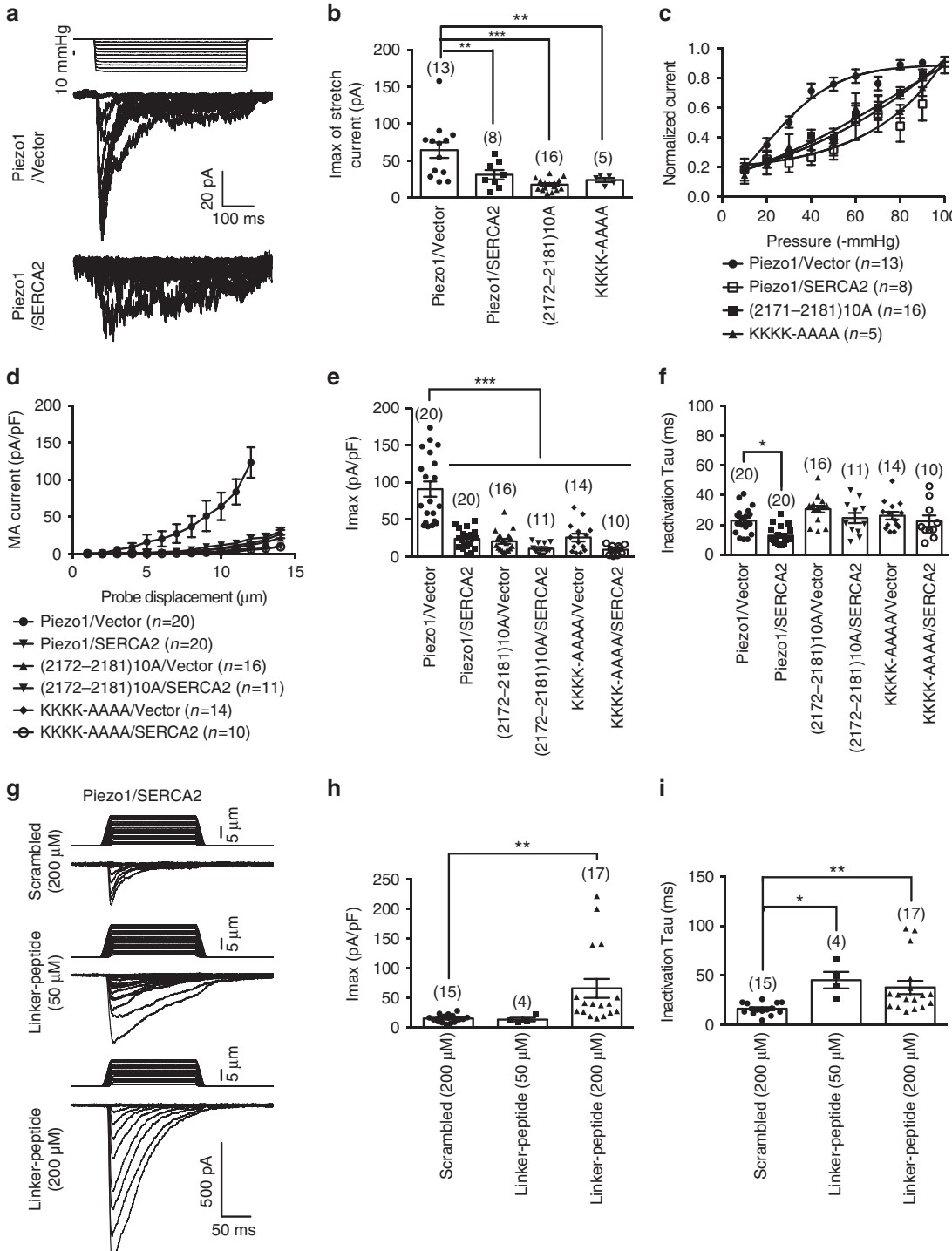

**Fig. 5** SERCA2 suppresses Piezo1 mechanosensitivity through the linker region. **a**, Representative stretch-induced currents recorded at -80 mV from HEK293T cells transfected with the indicated conditions. **b**, Scatter plots of the maximal stretch-induced currents. One-way ANOVA with multiple comparison test. **c**, Pressure-current relationships of the stretch-induced currents. The curves were fitted with a Boltzmann equation. The $P_{50}$ (pressure required for half maximal activation) for Piezo1/Vector-mediated current is -30.5 ± 1.7 mmHg. Given that the currents from the Piezo1/SERCA2, (2172–2181)10 A and KKKK-AAAA did not reach plateau, their $P_{50}$ value could not be accurately determined, but are estimated to be above -50 mmHg. Data shown as mean ± s.e.m. **d**, Relationship between poking-induced currents and the applied poking displacement recorded at -60 mv. **e** and **f**, Scatter plots of the maximal poking-induced currents (**e**) or inactivation tau (**f**) of the indicated transfections. One-way ANOVA with multiple comparison test. **g**, Representative current traces of poking-induced inward currents recorded at -60 mV from HEK293T cells transfected with Piezo1 and SERCA2 under the indicated conditions. **h** and **i**, Scatter plots of the maximal poking-induced currents (**h**) or inactivation tau (**i**) recorded from HEK293T cells transfected with Piezo1 and SERCA2 in the presence of the indicated peptides in the pipette solution. One-way ANOVA with multiple comparison test. Data shown as mean ± s.e.m., and the recorded cell number is labeled. $^{*}p < 0.05$, $^{**}p < 0.01$, $^{***}p < 0.001$

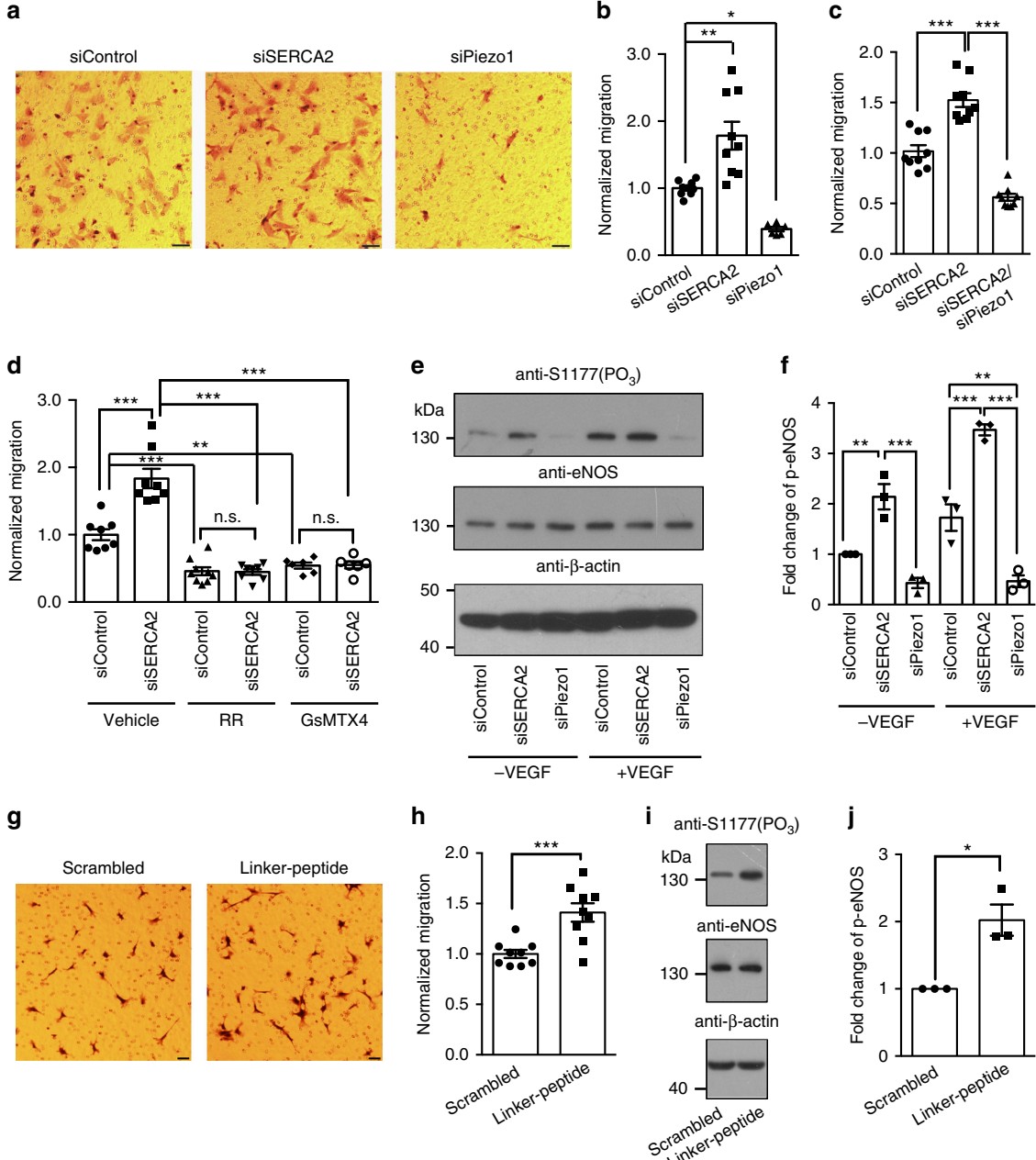

**Fig. 6** Regulation of Piezo1-dependent mechanotransduction processes by SERCA2 in HUVEC. **a** and **g**, Representative images showing the migrated HUVEC cells in the transwell assay. Scale bar, 50 μm (a) or 20 μm (g). **b**, **c**, **d**, and **h**, Scatter plots of the normalized migration ability of HUVEC under the indicated conditions. One-way ANOVA with Turkey's multiple comparison test (**b**, **c**, **d**) or unpaired student's t-test (**h**). **e** and **i**, Western blotting shows the phosphorylation at S1177 of the eNOS protein under the indicated conditions. **f** and **j**, Scatter plot of the normalized fold change of phosphorylated-eNOS under the indicated conditions. Data shown as mean ± s.e.m. $^*p < 0.05$, $^{**}p < 0.01$, $^{***}p < 0.001$

a relatively small endogenous poking-induced current ($1.1 \pm 0.1$ pA/pF). The current was significantly reduced when Piezo1 was knocked down ($0.46 \pm 0.04$ pA/pF) (Fig. 4f, g) or blocked with the mechanosensitive channel blocker GsMTx4[39,40] ($0.05 \pm 0.05$ pA/pF), but potentiated by Yoda1 ($6.4 \pm 2.1$ pA/pF) (Supplementary Fig. 3e). These observations suggest that the poking-induced currents in HUVEC were mediated by endogenously expressed Piezo1. When endogenous SERCA2 was knocked down, the poking-induced current was significantly enhanced to $6.9 \pm 2.2$ pA/pF (Fig. 4f, g). The efficiency of the siRNA-mediated knock-down of the Piezo1 and SERCA2 proteins in HUVEC is shown in Supplementary Fig. 7. Collectively, these data suggest that

SERCA2 suppresses endogenous Piezo1-mediated mechanosensitive currents in different cell types.

**SERCA2 suppresses Piezo1 through the linker region**. Given that the plasma membrane expression of Piezo1 was not affect by SERCA2 (Fig. 3a–d), we reasoned that the inhibition of Piezo1 currents by SERCA2 might be due to either suppression of Piezo1 mechanosensitivity or reduction of its unitary conductance. Analyzing the spontaneous single-channel activities from cells transfected with Piezo1/Vector or Piezo1/SERCA2 in the absence of applied external pressure revealed that SERCA2 did not affect

the unitary conductance of Piezo1 (Supplementary Fig. 4a–c). However, we found that the maximal stretch-induced current from cells transfected with Piezo1/SERCA2 (30.7 ± 6.3 pA) was significantly lower than that of Piezo1/Vector (64.1 ± 10.5 pA) (Fig. 5a, b), in line with the inhibitory effect of SERCA2 on poking-induced Piezo1 currents (Fig. 4a, b). Furthermore, SERCA2 co-expression caused a rightward shift of the pressure-current response curve of Piezo1 (Fig. 5c), indicating reduced mechanosensitivity of Piezo1. Collectively, these data suggest that the inhibition of Piezo1-mediated currents by SERCA2 is due to suppression of Piezo1 mechanosensitivity.

We next asked whether SERCA2 functionally modulates Piezo1 through the linker region. Consistent with their deficit in interacting with SERCA2, the Piezo1-(2172–2181)10A and Piezo1-KKKK-AAAA mutants did not show significant SERCA2-dependent inhibition of their poking-induced currents and fastened inactivation rate (Fig. 5d–f). Intriguingly, in line with the effect of the linker-peptide in disrupting the interaction between Piezo1 and SERCA2 (Fig. 2h, i), application of the linker-peptide to cells co-transfected with Piezo1 and SERCA2 led to a dose-dependent increase of the maximal poking-induced currents (Fig. 5g, h) and the associated inactivation Tau (Fig. 5i), reversing the inhibitory effect of SERCA2 on Piezo1 function. These data strongly suggest that the linker region of Piezo1 serve as the modulatory site for SERCA2.

Given that the linker region is highly conserved between Piezo1 and Piezo2 (Supplementary Fig. 5a), we investigated whether SERCA2 interacts with and modulates Piezo2. Indeed, similar to Piezo1, Piezo2 interacted with SERCA2 (Supplementary Fig. 5b). Furthermore, co-expression of SERCA2 drastically inhibited poking-evoked Piezo2 currents (Supplementary Fig. 5c–e). These data suggest that Piezo1 and Piezo2 share a similar modulatory mechanism by SERCA2.

**The linker is critical for mechanogating of Piezo1**. Despite their normal expression in the plasma membrane (Fig. 3e–h), the linker mutants themselves had lower Imax of stretch-induced currents (Fig. 5b) and a rightward shift of their pressure-current response curves (Fig. 5c), and drastically reduced poking-induced whole-cell currents (Fig. 5d–f). To rule out that the residual mechanosensitive currents of Piezo1-(2172–2181)10A- or Piezo1-KKKK-AAAA-transfected HEK293T cells were potentially mediated by endogenous Piezo1, we further examined their poking-induced currents in the Piezo1-KO-HEK293T cells where the endogenous Piezo1 gene is disrupted[41]. We observed consistent poking-evoked currents from Piezo1-(2172–2181)10A- or Piezo1-KKKK-AAAA-transfected Piezo1-KO-HEK293T cells, but not from vector-transfected cells (Supplementary Fig. 6a). Furthermore, the poking-induced currents of the mutant channels were significantly smaller than Piezo1-mediated currents (Supplementary Fig. 6). Single-channel analysis revealed that the unitary conductance of the two mutants was not different from that of Piezo1 (Supplementary Fig. 4d). Collectively, these data suggest that the linker mutants have severely impaired mechanosensitivity. Thus, likely by coupling the peripheral mechanotransduction-modules to the central ion-conducting pore, the linker plays a critical role in mechanogating of Piezo1.

**SERCA2 regulates Piezo1-dependent endothelial cell migration**. We next examined the functional importance of the SERCA2-mediated regulation of Piezo1 in affecting cellular mechanotransduction. Piezo1-mediated mechanotransduction has been shown to play critical roles in mediating the migration process of HUVEC[9], which might be required for proper development of blood vessels. Indeed, siRNA-mediated knockdown of

Piezo1 inhibited HUVEC migration as examined by the transwell assay (Fig. 6a, b). By contrast, knockdown of SERCA2 increased the cell migration (Fig. 6a, b). Importantly, the SERCA2 knockdown-induced effect on cell migration was inhibited either by simultaneously knocking down Piezo1 proteins (Fig. 6c) or functionally blocking Piezo1 channel activities using either the non-specific blocker ruthenium red (RR) or the relatively specific blocker GsMTx4 (Fig. 6d). The knockdown efficiency of SERCA2 and Piezo1 is shown in Supplementary Fig. 7.

Previous studies have suggested that the endothelial NO synthesis (eNOS) serves as a critical signaling transduction molecule involved in Piezo1-controlled cell migration[9]. We verified that knockdown of Piezo1 inhibited the phosphorylation of eNOS at the residue S1177 in HUVEC treated with or without the vascular endothelial growth factor (VEGF) (Fig. 6e, f). By contrast, knockdown of SERCA2 increased eNOS phosphorylation (Fig. 6e, f), in line with the observation that knockdown of SERCA2 resulted in an enhancement of Piezo1 activity and cell migration.

Lastly, we found that application of the linker-peptide to HUVEC cells led to increased cell migration (Fig. 6g, h) and eNOS phosphorylation (Fig. 6i, j), further demonstrating that the effect of SERCA2 in affecting HUVEC migration and eNOS phosphorylation is mediated through SERCA2-Piezo1 interaction. Taken together, our data reveal that modulation of Piezo1 activity by SERCA2 can be manifested into changes in Piezo1-mediated cellular mechanotransduction processes of crucial physiological significance.

## Discussion

The Piezo protein family, including Piezo1 and Piezo2, has been firmly established as the long-sought pore-forming subunits of mammalian mechanosensitive cation channels[4,5,27,28], and shown to play critical roles in various mechanotransduction processes examined to date[1]. Thus, it is pivotal to understand the mechanogating and regulatory mechanisms that enable Piezo channels to serve as sophisticated mechanotransducers for various mechanotransduction processes. Here, we have identified the SERCA protein family, exemplified by the widely expressed SERCA2 isoform, as interacting proteins of Piezo channels (Fig. 1), and revealed the critical role of the 14-residue-constituted intracellular linker region out of the 2547 residues of mouse Piezo1 for its mechanogating and regulation (Figs. 2, 5). Remarkably, the synthetic linker-peptide is effective in competing for Piezo1-SERCA2 interaction (Fig. 2), consequently modulating Piezo1-mediated mechanosensitive currents (Fig. 5) and cellular mechanotransduction processes such as cell migration (Fig. 6). Thus, our studies not only provide important insights into the mechanogating and regulatory mechanisms of Piezo channels, but also open a potential for therapeutic intervention of Piezo-derived human diseases by targeting the SERCA-Piezo interaction.

Mammalian Piezos are large transmembrane proteins that are composed of about 2500–2800 amino acids with large number of transmembane segments (TMs)[4]. Furthermore, they do not have sequence homology with other ion channels such as the 6-TM-based ion channels families, including the voltage-gated $K^+$, $Na^+$ and $Ca^{2+}$ channels and TRP channels. When reconstituted into lipid bilayers, purified Piezo1 proteins mediate spontaneous and membrane tension-induced cationic currents[5,42], demonstrating that they form intrinsically mechanosensitive cation channels. Despite its sequence complexity, our recent structural and functional characterizations reveal that Piezo1 trimerizes to form a three-bladed, propeller-like architecture comprising two distinct modules: the central ion-conducting pore-module formed by the

last-two-TM-containing C-terminal region (~ 2189–2547) and the peripheral propeller-like structures formed by the large N-terminal region (~ 1–2100)[27,28]. Based on the structural organizations and functional characterizations of Piezo1, we have proposed that Piezo1 might utilize its propeller-resembling structures as mechanotransduction-modules to mechanically gate the central pore-module[27,28]. This hypothesis would allow us to deduce the complicated Piezo channels into an analogous working model employed by voltage-gated channels that utilize the N-terminal voltage-sensing-module to gate the C-terminal pore-module, connected by a well-documented "S4-S5-linker"[29]. Remarkably, the linker mutants of Piezo1, including Piezo1-(2172–2181)10A and Piezo1-KKKK-AAAA, have drastically reduced mechanosensitive currents due to decreased mechanosensitivity (Fig. 5). These data suggest that the linker region plays a key role in transducing force-induced conformational changes of the N-terminal propeller-resembling structure into opening the pore, in analogous to the role of the S4-S5 linker of voltage-gated $K^+$ channels for electromechanical coupling of the voltage-sensing domain to the pore[29]. Thus, these results support the working model that Piezo1 might employ the peripheral propeller-structures as mechanotransduction-modules to gate the central pore-module[27,28].

Combining affinity pull-down of Piezo1 complex and mass spectrometry, we have identified SERCAs as interacting proteins of Piezo1 and Piezo2 (Fig. 1 and Supplementary Fig. 5). Importantly, we have obtained several lines of evidence to support that SERCA2 strategically binds to the linker for fine-tuning the mechanogating of Piezo1. First of all, the co-localization between Piezo1 and SERCA2 is more prominent near the PM than inside the cytosol (Fig. 1e, f), suggesting that the interaction might occur at the ER-PM junction. Thus, the cytoplasmic regions of the PM-localized Piezo1 and the ER-localized SERCA2 are likely to be involved in their interaction. Secondly, SERCA2 binds to the C-terminal fragments in accordance with the structural organization of the defined structural domains. Based on the structure of the fragment of 2171–2547, the linker and CTD are the only two intracellular exposing domains (Fig. 2a). The fragment of 2171–2483 that contains the linker but without CTD had the strongest interaction with SERCA2 (Fig. 2d, e). In sharp contrast, the fragment of 2186–2547 that contains the CTD but without the linker failed to interact with SERCA2 (Fig. 2d, e). These data demonstrate that the intracellular linker is essential for the C-terminal fragment of 2171–2547 to interact with SERCA2. Thirdly, mutating the linker in the full-length Piezo1 not only reduced SERCA2 interaction (Fig. 2f, g) but also abolished SERCA2-mediated inhibition of the mechanosensitive currents (Fig. 5d–f). Lastly, we show that the linker-peptide was able to disrupt the Piezo1-SERCA2 interaction (Fig. 2h, i), reverse SERCA2-mediated inhibition of Piezo1 mechanosensitive currents (Fig. 5g–i), and potentiate cell migration and eNOS phosphorylation (Fig. 6g–j), suggesting that the linker-peptide is able to compete for the Piezo1-SERCA2 interaction. Together, these data strongly suggest that SERCA2 might directly bind to the linker of Piezo1 for regulating its mechanosensitivity. Nevertheless, given that we have not been able to identify the reciprocal region in SERCA2 responsible for interacting with Piezo1, we could not totally exclude the possibility that the linker region might play an allosteric role in affecting the Piezo1-SERCA2 interaction. Since the linker region is rich in positively charged residues (7 out 14 residues), future studies will focus on addressing whether negatively charged residues in the cytoplasmic region of SERCA2 might be involved in Piezo1 interaction.

The finding that SERCA2 strategically binds to the linker for suppressing the mechanogating of Piezo1 is remarkable. To the best of our knowledge, despite the well-documented importance

of the S4-S5 linker for the 6-TM-containing ion channel families including voltage-gated channels and TRP channels, a direct protein targeting at this region has not yet been reported. Instead, ligand binding at the S4-S5 linker has been revealed for the capsaicin receptor TRPV1[43]. Thus, we reveal that protein interaction at the linker region represents an important regulatory mechanism for tuning the mechanogating properties of Piezo1, empowering its role in physiological mechanotransduction.

The SERCA family of proteins including SERCA1–3 is essential for recycling cytosolic $Ca^{2+}$ into the SR or ER $Ca^{2+}$ store, a process essential for maintaining $Ca^{2+}$ homeostasis in nearly all cell types including muscles and endothelial cells[31]. Thus, the SERCA-mediated regulation of Piezo channels might ubiquitously exist in Piezo-expressing cell types, and consequently has broad physiological implications. Indeed, we found that the endogenously expressed Piezo1 in N2A and HUVEC cells is functionally regulated by endogenous SERCA2 (Fig. 4). Furthermore, the SERCA2-mediated regulation of Piezo1 mechanosensitivity has a clear implication in regulating Piezo1-dependent mechanotransduction processes such as endothelial cell migration (Fig. 6). The expression of SERCA proteins can be altered by genetic mutations or under pathological conditions[31]. For instance, decreased expression of SERCA2 in keratinocytes caused by genetic mutations can lead to human Darier's disease[31], which is a rare autosomal dominant skin disorder characterized by loss adhesion between epidermal cells and abnormal keratinization. Keratinocytes have high expression of Piezo1[4]. Thus it would be interesting to determine whether the loss of SERCA2 inhibition of Piezo1 function might contribute to the disease phenotypes.

In summary, by identifying SERCAs as interacting proteins of Piezo channels and the linker as the key component involved in the mechanogating and regulation, our studies provide important insights into the mechanogating and regulatory mechanism and potential therapeutic intervention of this prototypic class of mammalian mechanosensitive cation channels.

## Methods

**cDNA clones and molecular cloning.** The mouse Piezo1 (mPiezo1) and mouse Piezo2 (mPiezo2) clones were generously provided by Dr. Ardem Patapoutian at the Scripps Research Institute. The Piezo1-Flag construct was generated by replacing the C-terminal GST-tag of the Piezo1-GST-ires-GFP or Piezo2-GST-ires-GFP construct with the Flag tag. The Flag-SERCA2 clone was a gift from Dr. Xun Huang at the Institute of Genetics and Developmental Biology, Chinese Academy of Sciences. All the mutations, truncations and other molecular cloning were conducted with the one step cloning kit according to the instruction manual (Vazyme Biotech)[28]. All constructs were verified by sequencing. The primers used for generating the constructs are listed in Supplementary Table 2.

**Cell culture and transfection.** Human embryonic kidney 293 T (HEK293T) cells were purchased from ATCC and cultured in Dulbecco's Modified Eagle Medium (DMEM) supplemented with 10% fetal bovine serum (FBS), 100 U ml⁻¹ penicillin and 100 µg ml⁻¹ streptomycin. Neuro-2A (N2A) cells were provided by Dr. Ardem Patapoutian at the Scripps Research Institute and cultured in Modified Eagle Medium (MEM) containing 10% FBS, non-essential amino acids, 1 mM sodium pyruvate, 100 U ml⁻¹ penicillin and 100 µg ml⁻¹ streptomycin. Human umbilical vein endothelial cells (HUVECs) were purchased from Allcells (Shanghai, China) and cultured using EGM-2 growth medium supplemented with EGM-2 bullet kit (Lonza) in the plates coated with 50 µg ml⁻¹ collagen-I (Sigma). HUVECs were used for the experiments for up to 8 passages. The cells were transfected using polyethylenimine (PEI) (Polysciences) or Lipofectamine 2000 (Invitrogen) according to the manufacturer's instructions.

**Antibodies.** The Piezo1 antibody was custom generated by Abgent (Suzhou, China). The procedure is summarized briefly as follows. The C-terminal extracellular region of mPiezo1 (amino acids 2218–2453) was expressed in bacteria and purified for immunization in rabbit, then the Piezo1 rabbit antibody was purified by antigen affinity chromatography. The antibody was used at concentrations of 1:500–1:2000 for western blotting. Other antibodies used for western blotting include rabbit anti-GST (Millipore, 1:3,000), mouse anti-SERCA2 (Thermo, MA3–910, 1:1,000), mouse anti-Flag (Sigma, clone M2, 1:3,000), mouse anti-eNOS

(BD Biosciences, 1:1000), mouse anti-p(S1177)-eNOS (BD Biosciences, 1:1,000), rabbit anti-β-actin (Cell Signaling Technology, 1:3,000).

**GST pull-down and co-immunoprecipitation**. Cell lysates derived from HEK293T transiently transfected with the indicated constructs were prepared using a buffer containing 25 mM NaPIPES, 140 mM NaCl, 1% CHAPS, 0.5% phosphatidylcholine (PC), 2.5 mM dithiothreitol (DTT), a cocktail of protease inhibitors (Roche) and PhosSTOP phosphatase inhibitors (Roche). For GST pull-down experiments, the glutathione magnetic beads (Pierce) were incubated with cell lysates at 4 °C for overnight. The beads were washed 5 times with a buffer containing 25 mM NaPIPES, 140 mM NaCl, 0.6% CHAPS, 0.14% PC, 2.5 mM DTT, a cocktail of protease inhibitors (Roche) and PhosSTOP phosphatase inhibitors (Roche) and boiled for 5 to 10 min in 1 × SDS loading buffer. The protein samples were separated by SDS-PAGE gels and then subjected to either silver staining according to the instruction manual (Sigma) or western blotting. For immunoprecipitation of heterologously expressed Piezo1-Flag, the anti-Flag M2 magnetic beads (Sigma) were used. For co-immunoprecipitation of endogenous Piezo1 and SERCA2, N2A cells were lysed in the Tris-HCl lysis buffer (10 mM Tris-HCl, pH7.5, 150 mM NaCl, 1 mM EDTA, 1% Triton X-100, 0.5% NP-40, a cocktail protease inhibitors and PhosSTOP phosphatase inhibitors) on ice for 1 h. The protein G magnetic beads (Cell Signaling Technology) were incubated with either IgG or the anti-SERCA2 antibody (Thermo, MA3-910, 1:100) at 4 °C for 2 h. Then the antibody-bound beads were incubated with the N2A cell lysates at 4 °C for overnight, and subsequently washed for 5 times with lysis buffer. The immunoprecipitated proteins were subjected to SDS-PAGE and western blotting analysis.

**Western blotting**. Protein samples derived from the GST pull-down, immunoprecipitation or HUVEC cell lysates (lysed in the Tris-HCl lysis buffer) were subjected to SDS-PAGE gels and electrophoresis separation. The proteins were transferred to 0.45 μm PVDF membranes (Millipore) for western blotting according to the procedure described previously[5]. In brief, the membrane was blocked by 5% non-fat milk (Bio-rad) in TBST buffer (TBS buffer with 0.1% Tween-20) and incubated with the primary antibodies for overnight. After 3 times of washing with the TBST buffer, the membrane was incubated with the peroxidase-conjugated anti-rabbit IgG secondary antibody (CST, 1:10,000) or anti-mouse IgG secondary antibody (pierce, 1:20,000) at room temperature for 1 h, followed by washing and detection using the enhanced chemiluminescence (ECL) detection kit (Pierce). All uncropped images of blots are shown in Supplementary Fig. 8.

**Cell surface biotinylation**. Cultured cells were washed with ice-cold DPBS with $Ca^{2+}$ and $Mg^{2+}$ (DPBSCM) (Beyotime Biotechnology) for 3 times, incubated with DPBSCM containing 0.5 mg/mL Sulfo-NHS-LC-Biotin (Pierce). After 45 min of incubation at 4 °C, the biotinylation reaction was stopped by replacing the Sulfo-NHS-LC-Biotin solution with 100 mM glycine solution. The cells were then lysed. 2% of the lysates were used as the whole-cell lysate samples and the remaining cell lysates were incubated for overnight with the streptavidin magnetic beads (Pierce) at 4 °C. After 5 times of washing, the precipitated sample was denatured and prepared for SDS-PAGE gel separation and western blotting.

**Mass spectrometry**. Purified protein samples were separated on the 8% SDS-PAGE gel and visualized by silver staining. Three protein bands present specifically in the Piezo1-GST sample but not in the GST control sample (near the 300 kDa, 250 kDa and 130 kDa molecular markers, respectively) were excised and subjected to mass spectrometry identification in the Protein Core Facility of Tsinghua University.

**RNA interference**. Knockdown of Piezo1 in N2A cells was achieved by infection of the lentivirus containing either control or mPiezo1 shRNAs. To produce the lentivirus, the PLL3.7 lentivirus vector containing the shRNA encoding sequence and the helper vectors (pMDLg/pRRE, pRSV-Rev and pCMV-VSV-G) were co-transfected into HEK293T cells using PEI. 72 to 96 h later, virus present in the culture medium were collected and filtered through a filter of 0.45 μm size (Millipore). The virus-containing medium was further concentrated by 200-fold using polyethylene glycol precipitation (Sigma). For infection of N2A cells, fresh MEM containing 10 μl virus solution and 8 μg mL$^{-1}$ hexadimethrine bromide (Sigma) was added into one well of 6-well pate. 72 h after infection, the cells were harvested for RNA isolation and subsequent quantitative Real-Time PCR (RT-PCR) for validating the knockdown efficiency. For knockdown experiments in HUVECs, the cells were transfected with 50 nM siRNA using Lipofectamine 2000 (Invitrogen) following the manufacturer's instructions. 4–6 h later, the medium was replaced with fresh EGM-2 medium. 72 h after transfection, the cells were harvested and cell lysates were subjected to western blotting for analyzing the knock-down efficiency. The primers for RT-PCR and sequences of shRNA or siRNA are listed in Supplementary Table 2.

**Generating the Piezo1-Flag knock-in N2A cell line**. All the procedures are followed as the protocol provided by Feng Zhang's laboratory[44]. In brief, single-guide

RNA (sgRNA) sequence was designed by the CRISPR Design Tool (http://crispr.mit.edu) and then a pair of complementary oligo DNA segments containing the sgRNA sequence were synthesized, annealed and inserted into the Cas9-gRNA expression plasmid pX330 (Addgene). The plasmid-based donor repair template was made with the pcDNA3.1 (-) plasmid (containing an ires-GFP reporter) by inserting a pair of mPiezo1 genome sequences (about 600 bp) flanking the site G2410 as homology arms and the inserted Flag tag sequence. N2A cells were transfected with the pX330 plasmid containing sgRNA sequence and the donor plasmid. 48 h after transfection, GFP positive cells were isolated and sub-cultured into 96-well plates (single cell per well) by fluorescence activated cell sorting (FACS). Then the grown cell clones were selected and insertion of the Flag-tag encoding sequence into the Piezo1 genome was detected by PCR and sequencing. The insert sequence of the donor plasmid are listed in Supplementary Table 3.

**Immunostaining**. Briefly, for live cell labeling, cells cultured on coverslips were incubated with the rabbit anti-Flag antibody (Sigma, clone M2, 1:100) at room temperature for 1 h. After washing with medium, cells were incubated with the Alexa Fluor 594 donkey-anti-rabbit IgG secondary antibody (Invitrogen, 1:200) at room temperature for 1 h. After washing, cells were fixed with 4% paraformaldehyde (PFA). For cell permeable staining, cells were fixed, permeabilized with 0.02% Triton X-100 and blocked with 3% bovine serum albumin (BSA) in 1 × PBS buffer. The cells were then incubated with the rabbit anti-Flag antibody (1:500) or mouse anti-SERCA2 (1:500) at room temperature for 1 h. After washing with the TBST buffer, cells were incubated with the Alexa Fluor 594 donkey-anti-rabbit IgG secondary antibody (Invitrogen, 1:500) or Alexa Fluor 488 donkey-anti-mouse IgG secondary antibody (Invitrogen, 1:500) at room temperature for 1 h. After washing, coverslips were mounted onto the glass slide for confocal imaging. All the imaging procedures were performed on the Nikon A1 confocal microscope (Nikon Instruments) or the DeltaVision Elite high resolution microscope (GE Healthcare Life Sciences) with 60 x oil objective (N.A. = 0.95) or 100 × oil objective (N.A. = 1.49) by using the 488 nm excitation wavelength and the 562 nm excitation wavelength. The images were analyzed using the Nikon NIS-Elements AR software or the SoftWoRx Explorer software (GE Healthcare Life Sciences). The co-localization of Piezo1 and SERCA2 either at the cell periphery or in the cytoplasm of the immunostained Piezo1-Flag knock-in N2A cells was calculated as Pearson's co-localization efficiency with the ImageJ software installed the JACoP plugin (National Institutes of Health). Multiple cells were analyzed for quantitative comparison as shown in Fig. 1f.

**Fura-2 single cell $Ca^{2+}$ imaging**. Flag-SERCA2-ires-GFP cDNA (0.5 μg) or Flag-SERCA2(C318R)-ires-GFP cDNA (0.5 μg)-transfected HEK293T cells were plated in 24-well plates, and subject to Fura-2 single cell $Ca^{2+}$ imaging about 36 h post transfection. Cells grown on the poly-D-lysine coated 8-mm round glass coverslips were washed with the buffer containing 1 × HBSS (1.3 mM $Ca^{2+}$) and 10 mM HEPES (pH 7.2), then incubated with 2.5 μM Fura-2-AM (Molecular Probes) and 0.05% Pluronic F-127 (Life technologies) for 30 min at room temperature, subsequently washed with the buffer. The coverslip was mounted into an inverted Nikon-Tie microscopy equipped with a CoolSNAP CCD camera and Lambda XL light box (Sutter Instrument), and GFP positive and negative cells were selected for measurement of the 340/380 ratio with a 20 × objective (N.A. = 0.75) using the MetaFluor Fluorescence Ratio Imaging software (Molecular Device).

**Whole-cell electrophysiology and mechanical stimulation**. The patch-clamp experiments were carried out with Axopatch 200B amplifier (Axon Instruments) or HEKA EPC10. For studying the regulatory effect of SERCA2 on Piezo1 WT or mutants, either Flag-SERCA2-ires-GFP/Piezo1-mRuby or SERCA2-ires-RFP/Piezo1-GST-ires-GFP were co-transfected for identifying co-expressing cells showing both GFP and mRuby or RFP signals. The observed mechanically activated currents were similar between the two transfection conditions, and therefore the data were combined in Fig. 5e–g.

For whole-cell patch clamp recordings, recording electrodes had a resistance of 2–3 MΩ when filled with internal solution composed of (in mM) 133 CsCl, 1 $CaCl_2$, 1 $MgCl_2$, 5 EGTA, 10 HEPES (pH 7.3 with CsOH), 4 MgATP and 0.4 $Na_2$GTP. The extracellular solution was composed of (in mM) 133 NaCl, 3 KCl, 2.5 $CaCl_2$, 1 $MgCl_2$, 10 HEPES (pH 7.3 with NaOH) and 10 glucose. All experiments were done at room temperature. Currents were sampled at 20 kHz, filtered at 2 kHz using Clampex 10.4 software (Axon Instruments) or Patchmaster software. Leak currents before mechanical stimulations were subtracted off-line from the current traces. Voltages were not corrected for a liquid junction potential (LJP).

Mechanical stimulation was delivered to the cell being recorded at an angle of 80° using a fire-polished glass pipette (tip diameter 3–4 μm) as described. Downward movement of the probe towards the cell was driven by a Clampex controlled piezo-electric crystal micro-stage (E625 LVPZT Controller/Amplifier; Physik Instrument). The probe had a velocity of 1 μm ms$^{-1}$ during the downward and upward motion and the stimulus was maintained for 150 ms. A series of mechanical steps in 1 μm increments was applied every 20 s and currents were recorded at a holding potential of -60 mV.

**Cell-attached electrophysiology**. Stretch-activated currents were recorded in the standard cell-attached patch clamp configuration. Currents were sampled at 20 kHz and filtered at 2 kHz. Pipette were filled with a solution consisting of (in mM) 130 NaCl, 5 KCl, 10 HEPES, 1 CaCl₂, 1 MgCl₂, 10 TEA-Cl (pH 7.3 with NaOH) and external solution used to zero the membrane potential consisted of (in mM) 140 KCl, 10 HEPES, 1 MgCl₂, 10 glucose (pH 7.3 with KOH). All experiments were done at room temperature. Membrane patches were stimulated with 500 ms negative pressure pulses through the recording electrode using Patchmaster controlled pressure clamp HSPC-1 device (ALA-scientific). Stretch-activated channels were recorded at a holding potential of -80 mV with pressure steps from 0 to -100 mm Hg (-10 mm Hg increments), and 4–11 recording traces were averaged per cell for analysis. Current-pressure relationships were fitted with a Boltzmann equation of the form: $I(P) = [1 + \exp(-(P - P_{50})/s)]^{-1}$, where I is the peak of stretch-activated current at a given pressure, P is the applied patch pressure (in mm Hg), $P_{50}$ is the pressure value that evoked a current value which is 50% of Imax, and s reflects the current sensitivity to pressure.

**Cell migration assay**. The cell migration assay was performed using the transwell permeable supports (8.0 μm pores, Corning). HUVECs transfected with siRNAs were starved with EGM-2 medium without FBS for 4–6 h, and then digested with 0.25% trypsin-EDTA (Gibco). Cells were re-suspended and sub-cultured into the transwell insert at a concentration of $5 \times 10^4$ cells per well in 100 μl EGM-2 medium containing 0.4% FBS. In the 24-well plates, the lower compartment was added with 500 μl EGM2 medium supplemented with 0.4% FBS and 25 ng ml$^{-1}$ VEGF (Peprotech). After incubation in a cell culture incubator supplemented with 0.5% $CO_2$ at 37 °C for 6–8 h, the insert was fixed by 4% PFA for 10 min at 37 °C. After washing with $1 \times$ PBS, the cells on the inside of each insert were swabbed gently and the underside of each insert was stained by 0.1% crystal violet (Amersco). The migrating cells were imaged using an Olympus IX73 light microscopy in 4–6 randomly chosen fields and quantified by counting cell numbers with ImageJ software (National Institutes of Health). The migration rates of different groups were normalized to the siControl group.

**Cell permeable peptides**. The peptide of the linker region (amino acids 2171–2185) of the mPiezo1 protein was synthesized and myristoylated at its N-terminus (myr-NH2-TEKKYPQPKGQKKKK-COOH) by GeneScript (Nanjng, China). The scrambled peptide was synthesized with the same composition and did not resemble any known protein (myr-NH2-KQKPKTKEKYKQKGT-COOH). For GST pull-down, the peptides were added to the cell lysates with a working concentration of 200 μM and incubated overnight. For the whole-cell patch clamp experiments, the peptides were pre-mixed in the internal solution (50 or 200 μM) and filled in the pipettes. For migration and western blotting with HUVECs, the peptides were added to the culture medium for at least 30 min with a concentration of 50 μM.

**Reagents**. siRNAs specifically targeting human SERCA2 (Product ID: SI02626673, SI02626680, SI03053337, SI03062710) were purchased from Qiagen. The siRNA targeting mouse SERCA2 was synthesized by Sigma. Scrambled siRNA and the siRNA targeting human Piezo1 were synthesized by GenePharma (Shanghai, China). Yoda1 was purchased from Maybrige. GsMTx4 was purchased from Tocris Bioscience. Other chemicals were purchased from Sigma or Ameresco.

**Data analysis**. All data are shown as mean ± SEM. Statistical significance was evaluated using either unpaired Student's *t*-test or one-way ANOVA for analyzing multiple samples.

**Data availability**. All relevant data are available from the corresponding author upon reasonable request.

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

## Acknowledgements

We thank Dr. Haiteng Deng and the Protein Core Facility of Tsinghua University for the mass spectrometry study, and the imaging core facility for helping with imaging studies, and Dr. Wayne Chen (University of Calgary) for critical reading the paper. This work was supported by the National Natural Science Foundation of China (31422027, 31371118, 31630090), the Ministry of Science and Technology (2015CB910102, 2016YFA0500402) and the Ministry of Education (Young Thousand Talent Program) to B.X.

## Author contributions

T.Z. did the biochemical and cell migration experiments; S.C. did the electrophysiology experiments; F.J. generated the knock-in cell line; Q.Z. generated constructs. B.X. directed the study and wrote the paper with help from the authors. All the authors read and edited the paper.

## Additional information

**Competing interests:** The authors declare no competing financial interests.

