## [Peer Review File · Nature Communications]

Reviewers' comments:

Reviewer #1 (Remarks to the Author):

In this manuscript, Zhang et al. identify a novel interacting partner of the mechanosensitive ion channel Piezo1 by a GST pull-down followed by mass spectrometry analysis of associated proteins. The interacting protein, SERCA2, inhibits Piezo1 activity as assayed by electrophysiology on both natively and heterologously expressed protein. The authors further propose that an intracellular linker region of Piezo1 is required for SERCA2 binding, and this same region must be intact for proper mechanogating of the channel. Finally, they show that SERCA2 modulates cell migration rates in HUVECs via a mechanism requiring Piezo1 activity.

While the identification of a novel Piezo1 binding partner that directly modulates activity is of high significance, my enthusiasm for the manuscript is greatly diminished due to the often poor experimental design, as well as over-interpretation of several key experiments. Really, all of the major points raised below absolutely need to be well-addressed in order to make this manuscript sound and convince me of the finding. Also, the manuscript has apparently been prepared for a shorter format (like Nature letter), and the overly short discussion should be extended to address some of the points:

Major points:

1) The authors do not investigate the specificity of the interaction between Piezo1 and SERCA2, although it came up as a hit in the screen. In supplementary table 1, the second-most identified associated protein is SERCA1- yet an interaction between SERCA1 and Piezo1 is not tested. This raises concerns of specificity, especially in SERCA1 knockdown-experiments. Can SERCA1 rescue? Can it bind? If not, what is its role? Relatedly, the authors do not test (or even comment on) a potential interaction between SERCA2 and Piezo2. Please comment on whether the linker region is conserved between the two isoforms, or preferably, test experimentally for an effect of SERCA2 on Piezo2.

2) The claim that SERCA2 binds to the linker region of Piezo1 is a gross over-interpretation of the data. While an intact linker region certainly seems necessary for the interaction, this could be an allosteric effect entirely and the binding site might be different. Additionally, the authors hypothesize an electrostatic effect between SERCA2 and Piezo1, based on the involvement of charged residues. There are 3D structures available for SERCA2- if a corresponding cluster of negative charges on SERCA2 also prevent the SERCA2-Piezo1 binding upon charge neutralization, this would provide much stronger evidence for an electrostatic binding at the Piezo1 linker. What is more it would address the important issue of specificity (major point 1).

3) The conclusion that the linker region plays a critical role in mechanogating is also an over-interpretation of the data. First, the authors must show that membrane expression of the mutant channels is identical to that of wild-type, and that the diminished currents are not merely a result of a trafficking defect. Second, the authors must show that these mutant channels are functional in a Piezo1-free background, as HEK293t cells express endogenous Piezo1 (Lukacs et al., 2015). It is particularly hard to interpret the effects (or lack thereof) of SERCA2 on these channels given that the currents are so small, and with the lack of negative controls (e.g., expression of empty vector), it is not entirely convincing that these are not endogenous Piezo1. Also, if this region is specifically proposed to be transducer of force-induced conformational changes into pore opening (p.10), then the key gating parameters of the mutants should be characterized as they were for channels co-expressed with SERCA2 (Figure S4, but see below, major point 4).

4) The proposed mechanism of the effect of SERCA2 on Piezo1 gating is flawed. Total total ionic current (I) is a function of the # of channels (N), the single channel conductance (g) and the open probability (P_o). Figure S3 shows that membrane expression levels (N) are unchanged by SERCA2, and Figure S4 shows that (g) and P_o are unchanged- so what causes the reduced current amplitudes? The reduced mechanosensitivity must result from a change in one of these parameters. Also, the acquisition and analysis of single channel events are not well-described.

What pressure was used to evoke openings? (This is particularly important, as interpretation of changes (or lack thereof) in P_o will be correlated to the location on the respective P50 curves of each construct). Do the reported n values represent sweeps, openings, or cells, and how many events were analyzed for each cell? Please also include representative open and closed time histograms and associated fits.

5) As presented, the FRET data are not convincing, as the changes in FRET efficiency are very small and the controls are not appropriate, as FRET signals depend on molecule numbers, which are different in each here compared condition. Unless proper controls can be generated I recommend removing these data from the manuscript. Also, what forester radius of proximity would be predicted for these efficiencies? Does this match the molecular arrangement of Piezo1 and SERCA1?

6) The manuscript would benefit from a brief discussion of the potential physiological significance of the SERCA2-Piezo1 interaction: in addition to HUVECs, in what other cell-types are the two proteins co-expressed? Under what physiological conditions is SERCA2 activity or expression up- or down-regulated, and how might those conditions relate to mechanosensitivity?

Minor points:

1) p.7: The current and standard deviations for Piezo1/SERCA2 (403.9 ± 65.2 pA) and Piezo1/SERCA2-C318R (403.9 ± 65.2 pA) are listed as the exact same values. If this is a typo, please fix.

2) p.8: Figure S5 (that endogenous Piezo1 currents can be modulated by SERCA2) is a key finding and should be a main figure.

3) Figure 1A: The gel does not seem to match the results described in the text: the band near 130 kDa actually appears in both groups, and the band next to SERCA2 (which looks closer to 150 kDa?) is hard to see. Also, what are the bands near 60 kDa that are solely present in the Piezo1 lane- were these also analyzed with mass spectrometry?

4) Figure 3b/h and 3c/i: The means for Piezo1/vector and Piezo1/SERCA2 I_{max} and τ look identical, but the n 's are different. Are these from different datasets? If so, why were they not pooled?

5) Figure 3e: Please report mean absolute current values for each construct (as was done in Figure 3b for poke).

6) Figure 3e-f: As currents from mutant constructs don't plateau, it is inappropriate to fit P50 values to these curves. Better state that " $P50 > xx$ mmHg".

7) Figure 4e-f: The authors conclude that knockdown of SERCA2 increases migration by increasing Piezo1 activity, but this could be an indirect effect. Do other manipulations that increase Piezo1 activity (e.g., Yoda1) also increase migration?

8) Figure 4e-f: Please describe (either in methods or figure legend) how the migration assay is quantified.

9) Figure S2: Given the extensive electrophysiological characterization of other mutant channels, the decision to characterize the Piezo1-Flag knock-in cells with Ca^{++} imaging and Yoda1 is odd- these currents should also be measured to provide a more quantitative report of the mutant channel activity.

Reviewer #2 (Remarks to the Author):

Review Zhang et al

In this paper Zhang and colleagues have identified and characterized an interaction between Piezo1 and the ER resident Ca²⁺ pump SERCA2. They provide biochemical evidence for this interaction and using a series of truncated constructs as well as Piezo1 deletion mutants they conclude that binding of the SERCA2 protein to the so called linker region of the Piezo1 channel leads to its inhibition. Finally, the authors provide evidence that this interaction in an endothelial cell line is relevant in regulating the ability of such cells to migrate in transwell migration assays. SERCA2 is not the only protein that has been proposed to provide an endogenous break on Piezo1 channel activity (e.g. see PC2 Peyronnet et al). The data in the manuscript is in general of high quality and the interaction between SERCA2 and Piezo1 appears convincing. The mechanism by which SERCA2 inhibits the channel is largely unaddressed in the MS. In particular the authors have not adequately addressed the possibility that the presence of SERCA2 regulates the number of channels reaching the membrane. This is especially critical as normally the Piezo1 channels and SERCA should not be in the same cellular compartment.

Major comments

I seriously doubt that peptides from SERCA were the only ones found when analyzing the extra band at around 130 (actually larger) in Figure 1 a. The authors should reveal all the peptides found in this experiment.

The pictures in Figure 1 e are very nice but it is not clear how consistent this localization was, impossible to tell without some quantification.

It is not clear how often the GST-pull downs were done. Are there at least three repeat experiments for each pull down? Qualitative statements about whether one pull down was more efficient than another are suspicious based on single experiments, I am thinking of the data in Figure 2 especially.

Figure 3 Mutations in the linker region lead to a striking loss of function. The evidence that these channels traffic properly to the membrane appears to be missing. Considering this it is hard to interpret a loss of modulation by SERCA in such mutants.

Figure 4 The currents measured in HUVECS do not look like PIEZO1 currents at all. They have different kinetics, but if their presence is dependent on Piezo1 it does not mean they are mediated by this channel. More evidence should be provided that these are PIEZO1 currents.

Supplementary Figure 3 The flag knockin cells have been tested only with the channel modulator Yoda. The tagged channel activity should be measured with electrophysiology. The immunofluorescence analysis of Flag tagged Piezo1 with and without SERCA is not really sufficient to address issues of trafficking. A more quantitative method would be to look at biotinylation of membrane localized Piezo1.

The poking induced current examples in Supplementary Figure 5 look very strange (no current deactivation). The examples do not appear to be typical when compared to the quantification.

Reviewer #3 (Remarks to the Author):

The manuscript by Dr. Xiao and co-workers entitled "A protein interaction mechanism for suppressing the mechanotransduction channel Piezo1" identifies an endoplasmic reticulum-localized Ca²⁺-ATPase (SERCA2) as a regulatory protein of Piezo1. Through elegant experiments, they show that SERCA2 binds to the linker between the pore domain and mechanotransduction domain of Piezo1 and modulates its mechanosensitivity. Therefore, in analogy to the role of the S4-S5 linker in voltage-sensitive ion channels, here authors propose that Piezo1 linker couples the conformational changes in the mechanosensing domain to the pore opening and SERCA2 binding might fine-tune the mechanogating process. Finally, they link the modulation of Piezo1 activity by SERCA2 to the endothelial cell migration.

The major claims of the paper are:

- Both heterologously and endogenously expressed Piezo1 and SERCA2 interacts with each other.
- Piezo1 and SERCA2 co-localize at the periphery of the cell but not in the cytosol or endoplasmic reticulum
- SERCA2 binds to the linker region of Piezo1. While it does not affect single channel properties, it affects the mechanosensitivity of the Piezo1.
- Modulation of Piezo1 activity by SERCA2 has implications in cellular mechanotransduction processes.

Novelty of the claims

- SERCA2 regulates Piezo1 by directly binding to it. So far, other regulators have been shown to affect the membrane curvature or stiffness.
- Linker region of Piezo1 plays a critical role in mechanogating of Piezo1 by functionally coupling the peripheral mechanotransduction module to the central ion conducting pore.
- Physiological implications of modulation of Piezo1 activity.

The findings will be of interest to scientists working in the fields of membrane proteins, especially mechanosensitive channels, pharmacology (e.g. Piezo1 linker as a drug target), and nanotechnology.

The work is convincing and well done. In each step, proper controls were included. Throughout the article, appropriate statistical tests used and all error bars and relevant statistical information were defined in the figure legends.

The article will influence thinking in the field. Especially, the analogy of the Piezo1 linker to the S4-S5 linker of voltage-sensitive channels will be of high interest.

My question and two comments are as follows:

1. In Figure 1a, there is a protein migrating near the 250 kDa marker, which is present only in Piezo1-GST incubated C2C12 cell extract, as SERCA2. Why did authors choose SERCA2 over this protein while searching for the interacting partners of Piezo1?

2. Figure legend 4d is missing

3. Authors use "mechanosensitive (MS) ion channels" at the beginning of the text. However, when explaining patch clamp results, they switch to the abbreviation "MA" (Mechanically-activated) current of these channels. Please use one of them throughout the text.

Responses to the reviewers' comments

Reviewer #1 (Remarks to the Author):

In this manuscript, Zhang et al. identify a novel interacting partner of the mechanosensitive ion channel Piezo1 by a GST pull-down followed by mass spectrometry analysis of associated proteins. The interacting protein, SERCA2, inhibits Piezo1 activity as assayed by electrophysiology on both natively and heterologously expressed protein. The authors further propose that an intracellular linker region of Piezo1 is required for SERCA2 binding, and this same region must be intact for proper mechanogating of the channel. Finally, they show that SERCA2 modulates cell migration rates in HUVECs via a mechanism requiring Piezo1 activity.

While the identification of a novel Piezo1 binding partner that directly modulates activity is of high significance, my enthusiasm for the manuscript is greatly diminished due to the often poor experimental design, as well as over-interpretation of several key experiments. Really, all of the major points raised below absolutely need to be well-addressed in order to make this manuscript sound and convince me of the finding. Also, the manuscript has apparently been prepared for a shorter format (like Nature letter), and the overly short discussion should be extended to address some of the points:

Major points:

1) The authors do not investigate the specificity of the interaction between Piezo1 and SERCA2, although it came up as a hit in the screen. In supplementary table 1, the second-most identified associated protein is SERCA1- yet an interaction between SERCA1 and Piezo1 is not tested. This raises concerns of specificity, especially in SERCA1 knockdown-experiments. Can SERCA1 rescue? Can it bind? If not, what is its role? Relatedly, the authors do not test (or even comment on) a potential interaction between SERCA2 and Piezo2. Please comment on whether the linker region is conserved between the two isoforms, or preferably, test experimentally for an effect of SERCA2 on Piezo2.

Response: We thank the reviewer for raising the constructive comments. We have carried out new experiments to address the interaction between all three isoforms of SERCA, including SERCA1-3 and Piezo1, as well as SERCA2 and Piezo2.

In addition to SERCA2, the two other isoforms of the SERCA family, SERCA1 (mainly expressed in skeletal muscles) and SERCA3 (expressed in a limited number of non-muscle cells), are also able to interact with Piezo1 (Supplementary Fig. 1, new data). Given the wide expression of SERCA2b, we focused on characterizing the interaction between SERCA2 and Piezo1.

We found that SERCA2 interacts with Piezo2 and suppresses its mechanosensitive currents (revised Supplementary Fig. 5, new data), similar to the regulatory effect of SERCA2 on Piezo1. Indeed, the linker region of Piezo1 that is required for interacting with SERCA2 is highly conserved between Piezo1 and Piezo2 (revised Supplementary Fig. 5a). These data suggest that Piezo1 and Piezo2 share a similar modulatory

mechanism by SERCA2.

2) The claim that SERCA2 binds to the linker region of Piezo1 is a gross over-interpretation of the data. While an intact linker region certainly seems necessary for the interaction, this could be an allosteric effect entirely and the binding site might be different. Additionally, the authors hypothesize an electrostatic effect between SERCA2 and Piezo1, based on the involvement of charged residues. There are 3D structures available for SERCA2- if a corresponding cluster of negative charges on SERCA2 also prevent the SERCA2-Piezo1 binding upon charge neutralization, this would provide much stronger evidence for an electrostatic binding at the Piezo1 linker. What is more it would address the important issue of specificity (major point 1).

Response: We agree with the reviewer for the critical comment that the linker region might play an allosteric role in affecting SERCA2 interaction with Piezo1. However, given that that the linker region is critically required for SERCA2 interaction to both the full-length Piezo1 and the structurally defined C-terminal fragments (revised Fig. 2), the most plausible hypothesis is that the linker serves as a direct binding site for SERCA2. To further validate this hypothesis, we synthesized the linker-peptide (2171-2185) and the scrambled control peptide with myristoylation at the N-terminal residue for allowing membrane penetration and then tested their effect in affecting Piezo1-SERCA2 interaction. Remarkably, the linker-peptide, but not the scrambled-peptide, significantly reduced the interaction between Piezo1 and SERCA2 (revised Fig. 2h and i, new data), indicating that the linker-peptide and Piezo1 compete for SERCA2 interaction. Furthermore, the linker-peptide functionally reversed the inhibitory effect of SERCA2 on Piezo1-mediated mechanosensitive currents (revised Fig. 5g-i, new data). Additionally, the linker-peptide potentiated HUVEC migration and eNOS phosphorylation (revised Fig. 6g-j, new data).

Together, we have several lines of evidence to support that SERCA2 binds to the linker for fine-tuning the mechanogating of Piezo1. First of all, the co-localization between endogenous Piezo1 and SERCA2 is more prominent near the plasma membrane than inside the cytosol (revised Fig. 1e, f), suggesting that the interaction might occur at the ER-PM junction. Thus, the cytoplasmic regions of the PM-localized Piezo1 and the ER-localized SERCA2 are likely to be involved in their interaction. Secondly, SERCA2 binds to the C-terminal fragments in accordance with the structural organization of the defined structural domains. Based on the structure of the fragment of 2171-2547, the linker and CTD are the only two intracellular domains (revised Fig. 2a). The fragment of 2171-2483 that contains the linker but without CTD had the strongest interaction with SERCA2 (revised Fig. 2d, e). In sharp contrast, the fragment of 2186-2547 that contains the CTD but without the linker failed to interact with SERCA2 (revised Fig. 2d, e). These data demonstrate that the intracellular linker is essential for the C-terminal fragment of 2171-2547 to interact with SERCA2. Thirdly, mutating the linker in the full-length Piezo1 not only reduced SERCA2 interaction (revised Fig. 2f, g) but also abolished SERCA2-mediated inhibition of the mechanosensitive currents (revised Fig. 5d - f). Lastly, we show that the linker-peptide was able to disrupt the Piezo1-SERCA2 interaction (revised

Fig. 2h, i, new data), reverse SERCA2-mediated inhibition of Piezo1 mechanosensitive currents (revised Fig. 5g-i, new data), and potentiate cell migration and eNOS phosphorylation (revised Fig. 6g-j, new data), suggesting that the linker-peptide is able to compete for the Piezo1-SERCA2 interaction. Together, these data strongly suggest that SERCA2 binds to the linker of Piezo1 for regulating its mechanosensitivity. Thus, our studies not only provide important insights into the mechanogating and regulation mechanisms of Piezo channels, but also open a novel potential for therapeutic intervention of Piezo-derived human diseases by targeting the SERCA-Piezo interaction.

The linker region is rich in positively charged residues, leading us to hypothesize an electrostatic effect between SERCA2 and Piezo1 for their interaction. Following the reviewer's constructive comment, we examined the 3D structure of SERCA1 and found a cluster of 4 acidic residues located at the top of the cytoplasmic region of SERCA1 (Rebuttal Fig. 1a). These residues are conserved among SERCA1-3 (Rebuttal Fig. 1b). We then generated the SERCA2 mutant by mutating the 4 acidic residues to alanine. Unfortunately, the mutant has severely reduced expression compared to wild type SERCA2 (Rebuttal Fig. 1c), preventing us from further examining the effect of mutating these acidic residues in Piezo1-SERCA2 interaction.

Thus, despite that we have strong evidence in supporting that SERCA2 binds to the linker region of Piezo1 as discussed above, in the discussion of the revised manuscript, we state that “given that we have not been able to identify the reciprocal region in SERCA2 responsible for interacting with Piezo1, we could not totally exclude the possibility that the linker region might play an allosteric role in affecting the Piezo1-SERCA2 interaction. Since the linker region is rich in positively charged residues (7 out of 14 residues), future studies will focus on addressing whether negatively charged residues in the cytoplasmic region of SERCA2 might be involved in Piezo1 interaction.”

3) The conclusion that the linker region plays a critical role in mechanogating is also an over-interpretation of the data. First, the authors must show that membrane expression of the mutant channels is identical to that of wild-type, and that the diminished currents are not merely a result of a trafficking defect. Second, the authors must show that these mutant channels are functional in a Piezo1-free background, as HEK293t cells express

endogenous Piezo1 (Lukacs et al., 2015). It is particularly hard to interpret the effects (or lack thereof) of SERCA2 on these channels given that the currents are so small, and with the lack of negative controls (e.g., expression of empty vector), it is not entirely convincing that these are not endogenous Piezo1. Also, if this region is specifically proposed to be transducer of force-induced conformational changes into pore opening (p.10), then the key gating parameters of the mutants should be characterized as they were for channels co-expressed with SERCA2 (Figure S4, but see below, major point 4). Response: We thank the reviewer for the constructive comments. We have carried out new experiments to show that the plasma membrane expression of the mutant channels is similar to that of wild-type Piezo1. To examine the localization of Piezo1 in plasma membrane, we inserted a Flag-tag after A2419 located in the extracellular CED into the Piezo1-GFP, 2172-2181(10A)-GFP and KKKK/AAAA-GFP fusion constructs (Piezo1-A2419Flag-GFP, 2172-2181(10A)-A2419Flag-GFP and KKKK/AAAA-A2419Flag-GFP, respectively) then carried out live immunostaining of the Flag-tag from HEK293T cells transfected with the constructs without permeabilizing the membrane. The GFP images were taken as control for the expression of the fusion proteins. As shown in **the revised Fig. 3a, e (new data)**, the anti-Flag immunofluorescent signal was specifically detected at the periphery of cells transfected with the Flagged constructs, but not from Piezo1-GFP-expressing cells. Quantitative analysis of the fluorescence intensity ratio of the anti-Flag signal over the GFP signal revealed that either co-expression of SERCA2 or mutating the linker region did not affect the plasma membrane expression of Piezo1 (**revised Fig. 3b, f, new data**).

To validate the live immunostaining results, we further carried out cell surface protein biotinylation assay. Western blotting of the Piezo1-GST, 2172-2181(10A)-GST and KKKK/AAAA-GST protein in the whole-cell lysate revealed that the overall expression of the Piezo1 proteins was neither affected by co-expression of SERCA2 (**revised Fig. 3c, new data**) nor by the linker mutations (**revised Fig. 3g, new data**). Furthermore, western blotting of the biotinylated protein samples pulled-down via streptavidin-beads shows similar biotinylated Piezo1-GST proteins with or without SERCA2 co-expression (**revised Fig. 3c, d, new data**) or between wild type and the linker mutants of Piezo1 (**revised Fig. 3g, h, new data**). These results are consistent with the live immunofluorescent studies showing that co-expression of SERCA2 or mutating the linker region did not affect the plasma membrane expression of Piezo1. Collectively, these data suggest that the inhibition of Piezo1 current by SERCA2 interaction or mutating the linker region was not due to traffic defect.

Given the reviewer's concern that the residual mechanosensitive currents of Piezo1-(2172-2181)10A- or Piezo1-KKKK-AAAA-transfected HEK293T cells might potentially be mediated by endogenous Piezo1, we have characterized the mutants in the Piezo1-KO-HEK293T cells where the encoding sequence of the Piezo1 gene is disrupted. We observed consistent poking-evoked currents from Piezo1-(2172-2181)10A- or Piezo1-KKKK-AAAA-transfected Piezo1-KO-HEK293T cells, but not from vector-transfected cells (**Supplementary Fig. 6a**). Furthermore, the mutant-mediated poking-induced

currents were significantly smaller than Piezo1-mediated currents (Supplementary Fig. 6), consistent with the results obtained with the use of HEK293T cells as the heterologous expression system. Thus, our data suggest that the linker mutants are intrinsically impaired for generating mechanosensitive currents.

We have characterized single-channel analysis in the absence of externally applied negative pressure and revealed that the unitary conductance of the two mutants was not different from that of Piezo1 (Supplementary Fig. 4d). Furthermore, the mutant channels had similar plasma membrane expression as Piezo1 (Fig. 3e-h). The reduced mechanosensitive currents of the linker mutants are due to their reduced mechanosensitivity (Fig. 5a-c). Thus, likely by coupling the peripheral mechanotransduction-modules to the central ion-conducting pore, the linker plays a critical role in mechanogating of Piezo1.

We have included these new data and discussion in the revised manuscript.

4) The proposed mechanism of the effect of SERCA2 on Piezo1 gating is flawed. Total ionic current (I) is a function of the # of channels (N), the single channel conductance (g) and the open probability (P_o). Figure S3 shows that membrane expression levels (N) are unchanged by SERCA2, and Figure S4 shows that (g) and P_o are unchanged- so what causes the reduced current amplitudes? The reduced mechanosensitivity must result from a change in one of these parameters. Also, the acquisition and analysis of single channel events are not well-described. What pressure was used to evoke openings? (This is particularly important, as interpretation of changes (or lack thereof) in P_o will be correlated to the location on the respective P50 curves of each construct). Do the reported n values represent sweeps, openings, or cells, and how many events were analyzed for each cell? Please also include representative open and closed time histograms and associated fits.

Response: We apologize for the confusion. For the data shown in the Supplementary Figure 4 of the original manuscript, the single channel conductance (g) and P_o were calculated from spontaneous channel activity without application of external mechanical stimulation. Applying negative pressures evoked macroscopic currents as shown in the revised Fig. 5a. At both -10 mmHg and -20 mmHg, the stretch-induced currents between Piezo1/Vector- and Piezo1/SERCA2-transfected cells were not significantly different. While from -30 mmHg to -90 mmHg, the stretch-induced currents from Piezo1/SERCA2-transfected cells were significantly smaller than that from Piezo1/Vector-transfected cells. Under these high negative pressure, the macroscopic currents prevented accurate single-channel analysis. Thus, showing the unchanged P_o in the absence of negative pressure was misleading.

Given that the plasma membrane expression of Piezo1 was not affect by SERCA2 (Fig. 3a - d), we reasoned that the inhibition of Piezo1 currents by SERCA2 might be due to either suppression of Piezo1 mechanosensitivity or reduction of its unitary conductance. Therefore, in the revised Supplementary Figure 4, we focused on showing that the

single-channel conductance of Piezo1 was not affected by co-expression of SERCA2 and removed the Po and open time analysis. Instead, we found that the inhibition of Piezo1-mediated currents by SERCA2 is due to suppression of Piezo1 mechanosensitivity (revised Fig. 5a-c).

The reported n values represent analyzed cells. 150 to 1400 or 40 to 1800 events were analyzed for individual Piezo1/Vector- (5 cells) or Piezo1/SERCA2-transfected (4 cells) cells, respectively. We have included these information in the legend for Supplementary Fig. 4b.

5) As presented, the FRET data are not convincing, as the changes in FRET efficiency are very small and the controls are not appropriate, as FRET signals depend on molecule numbers, which are different in each here compared condition. Unless proper controls can be generated I recommend removing these data from the manuscript. Also, what forester radius of proximity would be predicted for these efficiencies? Does this match the molecular arrangement of Piezo1 and SERCA1?

Response: Following the reviewer's suggestion, we have removed the data.

6) The manuscript would benefit from a brief discussion of the potential physiological significance of the SERCA2-Piezo1 interaction: in addition to HUVECs, in what other cell-types are the two proteins co-expressed? Under what physiological conditions is SERCA2 activity or expression up- or down-regulated, and how might those conditions relate to mechanosensitivity?

Response: We thank the reviewer for the suggestion. In the revised manuscript, we have included the discussion section. We have discussed the potential physiological significance of the SERCA2-Piezo1 interaction as the following:

“The SERCA family of proteins including SERCA1-3 is essential for recycling cytosolic Ca^{2+} into the SR or ER Ca^{2+} store, a process critical for maintaining Ca^{2+} homeostasis in nearly all cell types including muscles and endothelial cells. Thus, the SERCA-mediated regulation of Piezo channels might ubiquitously exist in Piezo-expressing cell types, and consequently has broad physiological implications. Indeed, we found that the endogenously expressed Piezo1 in N2A and HUVEC cells is functionally regulated by endogenous SERCA2 (Fig. 4). Furthermore, the SERCA2-mediated regulation of Piezo1 mechanosensitivity has a clear implication in regulating Piezo1-dependent mechanotransduction processes such as endothelial cell migration (Fig. 6). The expression of SERCA proteins can be altered by genetic mutations or under pathological conditions. For instance, decreased expression of SERCA2 in keratinocytes caused by genetic mutations can lead to human Darier's disease, which is a rare autosomal dominant skin disorder characterized by loss adhesion between epidermal cells and abnormal keratinization. Keratinocytes have high expression of Piezo1. Thus it would be interesting to determine whether the loss of SERCA2 inhibition of Piezo1 function might contribute to the disease phenotypes.”

Minor points:

1) p.7: The current and standard deviations for Piezo1/SERCA2 (403.9 ± 65.2 pA) and Piezo1/SERCA2-C318R (403.9 ± 65.2 pA) are listed as the exact same values. If this is a typo, please fix.

Response: We apologize for the typo. The I_{max} for Piezo1/SERCA2 and Piezo1/SERCA2-C318R is 403.9 ± 65.2 pA vs 400.3 ± 78.3 pA, respectively.

2) p.8: Figure S5 (that endogenous Piezo1 currents can be modulated by SERCA2) is a key finding and should be a main figure.

Response: We thank the reviewer for the suggestion. In the revised manuscript, we have included the data in the revised main Fig. 4.

3) Figure 1A: The gel does not seem to match the results described in the text: the band near 130 kDa actually appears in both groups, and the band next to SERCA2 (which looks closer to 150 kDa?) is hard to see. Also, what are the bands near 60 kDa that are solely present in the Piezo1 lane- were these also analyzed with mass spectrometry?

Response: We thank the reviewer for pointing this out. After examining the original data, we found that the position of the 130 kD-molecular marker was not correctly positioned during the figure preparation process. The 130 kD-molecular marker should be near the SERCA2 band. We have corrected this in the revised Fig. 1a. We apologize for the error.

As for the bands near 60 kDa, since we observed weaker protein bands in the GST-control sample at the exact same positions, we did not analyze the bands with mass spectrometry.

4) Figure 3b/h and 3c/i: The means for Piezo1/vector and Piezo1/SERCA2 I_{max} and τ look identical, but the n's are different. Are these from different datasets? If so, why were they not pooled?

Response: For the previous Fig. 3b, c (revised Fig. 4b, c), the combination of Flag-SERCA2-ires-GFP/Piezo1-mCherry was used for identifying co-transfection of SERCA2 and Piezo1. For the previous Fig. 3h, i (revised Fig. 5e, f), the combination of SERCA2-ires-RFP/Piezo1 (wt or mutant)-pp-GST-ires-GFP were co-transfected. We found that the observed poking-induced currents were similar between the two transfection conditions, and therefore the data were pooled in the revised Fig. 5d - f.

5) Figure 3e: Please report mean absolute current values for each construct (as was done in Figure 3b for poke).

Response: We thank the reviewer for the suggestion. We have included the I_{max} of stretch-induced current in the revised Fig. 5b. Consistent with the poking-induced I_{max} , the stretch-induced I_{max} is significantly reduced upon either co-expression of Piezo1 or mutating the linker.

6) Figure 3e-f: As currents from mutant constructs don't plateau, it is inappropriate to fit P50 values to these curves. Better state that "P50 > xx mmHg".

Response: We thank the reviewer for the suggestion and removed the panel (original Fig. 3f) showing the P_{50} values. We specified in the legend of the revised Fig. 5c that “c, Pressure-current relationships of the stretch-induced currents. The curves were fitted with a Boltzmann equation. The P_{50} for Piezo1/Vector-mediated current is -30.5 ± 1.7 mmHg. Given that the currents from the Piezo1/SERCA2, (2172-2181)10A and KKKK-AAAA did not reach plateau, their P_{50} value could not be accurately determined, but are estimated to be above -50 mmHg.”

7) Figure 4e-f: The authors conclude that knockdown of SERCA2 increases migration by increasing Piezo1 activity, but this could be an indirect effect. Do other manipulations that increase Piezo1 activity (e.g., Yoda1) also increase migration?

Response: We have found that the linker-peptide increased the migration and eNOS phosphorylation as shown in the revised Fig. 6g-j (new data), providing strong evidence that the increased migration upon SERCA2 knockdown is due to release inhibition of Piezo1.

8) Figure 4e-f: Please describe (either in methods or figure legend) how the migration assay is quantified.

Response: We thank the reviewer for the comment. In the Method section of the revised manuscript, we have described the migration assay. “The migrating cells were imaged using an Olympus IX73 light microscopy in 4-6 randomly chosen fields and quantified by counting cell numbers with ImageJ software (National Institutes of Health). The migration rates of different groups were normalized to the siControl group.”

9) Figure S2: Given the extensive electrophysiological characterization of other mutant channels, the decision to characterize the Piezo1-Flag knock-in cells with Ca^{++} imaging and Yoda1 is odd-these currents should also be measured to provide a more quantitative report of the mutant channel activity.

Response: We have done electrophysiological characterization of the Piezo1-Flag knock-in cells and found that the poking-induced Piezo1-dependent currents were similar between wild-type N2A and Piezo1-Flag knock-in N2A cells (revised Supplementary Fig. 2f-h, new data).

Reviewer #2 (Remarks to the Author):

Review Zhang et al

In this paper Zhang and colleagues have identified and characterized an interaction between Piezo1 and the ER resident Ca^{2+} pump SERCA2. They provide biochemical evidence for this interaction and using a series of truncated constructs as well as Piezo1 deletion mutants they conclude that binding of the SERCA2 protein to the so called linker region of the Piezo1 channel leads to its inhibition. Finally, the authors provide evidence that this interaction in an endothelial cell line is relevant in regulating the ability of such cells to migrate in transwell migration assays. SERCA2 is not the only protein that has been proposed to provide an endogenous break on Piezo1 channel activity (e.g. see PC2 Peyronnet et al). The data in the manuscript is in general of high quality and the interaction between SERCA2 and Piezo1 appears convincing. The mechanism by which SERCA2 inhibits the channel is largely unaddressed in the MS. In particular the authors have not adequately addressed the possibility that the presence of SERCA2 regulates the number of channels reaching the membrane. This is especially critical as normally the Piezo1 channels and SERCA should not be in the same cellular compartment.

Response: We thank the reviewer for the positive and constructive comments about our paper. We have carried out new experiments to address the reviewer's major concern whether SERCA2 regulates the number of channels reaching the membrane.

Major comments:

1. I seriously doubt that peptides from SERCA were the only ones found when analyzing the extra band at around 130 (actually larger) in Figure 1 a. The authors should reveal all the peptides found in this experiment.

Response: We thank the reviewer for the comments. After examining the original data, we found that the position of the 130 kDa-molecular marker was not correctly positioned during the figure preparation process. The 130 kDa-marker should be near the SERCA2 band. We have corrected this in the **revised Fig. 1a**. We apologize for the error.

We have revealed all the peptides identified in the 130 kDa-band in the **Supplementary Excel File**.

2. The pictures in Figure 1 e are very nice but it is not clear how consistent this localization was, impossible to tell without some quantification.

Response: We thank the reviewer for the comment. Following the reviewer's suggestion, the co-localization of Piezo1 and SERCA2 either at the cell periphery or in the cytoplasm of the immunostained Piezo1-Flag Knock-in N2A cells was calculated as Pearson's co-localization efficiency with the ImageJ software installed JACoP plugin (National Institutes of Health). Multiple cells were analyzed for quantitative comparison as shown in the **revised Fig. 1f (new data)**.

3. It is not clear how often the GST-pull downs were done. Are there at least three

repeat experiments for each pull down? Qualitative statements about whether one pull down was more efficient than another are suspicious based on single experiments, I am thinking of the data in Figure 2 especially.

Response: We thank the reviewer for the constructive comments. The experiments were repeated at least 3 times. We have included the quantified results in the revised Fig. 2c, e, g, I (new analysis).

4. Figure 3 Mutations in the linker region lead to a striking loss of function. The evidence that these channels traffic properly to the membrane appears to be missing. Considering this it is hard to interpret a loss of modulation by SERCA in such mutants.

Response: We thank the reviewer for the constructive comments. We have carried out new experiments to show that the plasma membrane expression of the mutant channels is identical to that of wild-type Piezo1. To examine the localization of Piezo1 in plasma membrane, we inserted a Flag-tag after A2419 located in the extracellular CED23 into the Piezo1-GFP, 2172-2181(10A)-GFP and KKKK/AAAA-GFP fusion constructs (Piezo1-A2419Flag-GFP, 2172-2181(10A)-A2419Flag-GFP and KKKK/AAAA-A2419Flag-GFP, respectively) then carried out live immunostaining of the Flag-tag from HEK293T cells transfected with the constructs without permeabilizing the membrane. The GFP images were taken as control for the expression of the fusion proteins. As shown in the revised Fig. 3a, e (new data), the anti-Flag immunofluorescent signal was specifically detected at the periphery of cells transfected the Flagged constructs, but not from Piezo1-GFP-expressing cells. Quantitative analysis of the fluorescence intensity ratio of the anti-Flag signal over the GFP signal revealed that either co-expression of SERCA2 or mutating the linker region did not affect the plasma membrane expression of Piezo1 (revised Fig. 3b, f, new data).

To validate the live immunostaining results, we have further carried out cell surface protein biotinylation assay. Western blotting of the Piezo1-GST, 2172-2181(10A)-GST and KKKK/AAAA-GST protein in the whole-cell lysate revealed that the overall expression of the Piezo1 proteins was neither affected by co-expression of SERCA2 (revised Fig. 3c, new data) nor by the linker mutations (revised Fig. 3g, new data). Furthermore, western blotting of the biotinylated protein samples pulled-down via streptavidin-beads shows similar biotinylated Piezo1-GST proteins with or without SERCA2 co-expression (revised Fig. 3c, d, new data) or between wild type and the linker mutants of Piezo1 (revised Fig. 3g, h, new data). These results are consistent with the live immunofluorescent studies showing that co-expression of SERCA2 or mutating the linker region did not affect the plasma membrane expression of Piezo1. Collectively, these data suggest that the inhibition of Piezo1 current by SERCA2 interaction or mutating the linker region was not due to traffic defect.

5. Figure 4 The currents measured in HUVECS do not look like PIEZO1 currents at all. They have different kinetics, but if their presence is dependent on Piezo1 it does not mean they are mediated by this channel. More evidence should be presented.

Response: We have shown that the poking-induced currents in HUVECs are dependent

on Piezo1 proteins as knockdown of Piezo1 significantly reduced the currents (**revised Fig. 4f, g**). Indeed, the kinetics of the current recorded in HUVECS is different from heterologously expressed Piezo1 in HEK293T cells. The endogenous currents show slowed inactivation. Similar endogenous Piezo1-dependent currents have also been reported in renal tubular epithelium cells (Peronnet et al., EMBO 2013). Thus, we think that the poking-induced currents in HUVECs were mediated by Piezo1. To further support this, we have carried out pharmacological experiments to show that the currents were potentiated by the Piezo1 chemical activator Yoda1 and blocked by the toxin blocker GsMTX4 (**revised Supplementary Fig. 3e, new data**).

6. Supplementary Figure 3 The flag knockin cells have been tested only with the channel modulator Yoda. The tagged channel activity should be measured with electrophysiology. The immunofluorescence analysis of Flag tagged Piezo1 with and without SERCA is not really sufficient to address issues of trafficking. A more quantitative method would be to look at biotinylation of membrane localized Piezo1.

Response: We thank the reviewer for the constructive comments. We have done electrophysiological characterization of the Piezo1-Flag knock-in cells and found that the poking-induced Piezo1-dependent currents were similar between wild-type N2A and Piezo1-Flag knock-in N2A cells (**revised Supplementary Fig. 2f-h, new data**).

Please refer to our response to the comment 4 of this reviewer, we have carried out both live immunofluorescent analysis and cell surface biotinylation assay for examining the plasma membrane expression of Piezo1 upon co-expression with SERCA2 or mutating the linker region (**revised Fig. 3, new data**). Our data suggest that the plasma membrane expression of Piezo1 was not affected.

7. The poking induced current examples in Supplementary Figure 5 look very strange (no current deactivation). The examples do not appear to be typical when compared to the quantification.

Response: We thank the reviewer for the comment. In the **revised Fig. 3d**, we have replaced the current traces recorded from another cell, which show current deactivation.

Reviewer #3 (Remarks to the Author):

The manuscript by Dr. Xiao and co-workers entitled “A protein interaction mechanism for suppressing the mechanotransduction channel Piezo1” identifies an endoplasmic reticulum-localized Ca²⁺ATPase (SERCA2) as a regulatory protein of Piezo1. Through elegant experiments, they show that SERCA2 binds to the linker between the pore domain and mechanotransduction domain of Piezo1 and modulates its mechanosensitivity. Therefore, in analogy to the role of the S4-S5 linker in voltage-sensitive ion channels, here authors propose that Piezo1 linker couples the conformational changes in the mechanosensing domain to the pore opening and SERCA2 binding might fine-tune the mechanogating process. Finally, they link the modulation of Piezo1 activity by SERCA2 to the endothelial cell migration.

The major claims of the paper are:

- Both heterologously and endogenously expressed Piezo1 and SERCA2 interacts with each other.
- Piezo1 and SERCA2 co-localize at the periphery of the cell but not in the cytosol or endoplasmic reticulum
- SERCA2 binds to the linker region of Piezo1. While it does not affect single channel properties, it affects the mechanosensitivity of the Piezo1.
- Modulation of Piezo1 activity by SERCA2 has implications in cellular mechanotransduction processes.

Novelty of the claims

- SERCA2 regulates Piezo1 by directly binding to it. So far, other regulators have been shown to affect the membrane curvature or stiffness.
- Linker region of Piezo1 plays a critical role in mechanogating of Piezo1 by functionally coupling the peripheral mechanotransduction module to the central ion conducting pore.
- Physiological implications of modulation of Piezo1 activity.

The findings will be of interest to scientists working in the fields of membrane proteins, especially mechanosensitive channels, pharmacology (e.g. Piezo1 linker as a drug target), and nanotechnology.

The work is convincing and well done. In each step, proper controls were included. Throughout the article, appropriate statistical tests used and all error bars and relevant statistical information were defined in the figure legends.

The article will influence thinking in the field. Especially, the analogy of the Piezo1 linker to the S4-S5 linker of voltage-sensitive channels will be of high interest.

Response: We thank the reviewer for the highly positive comments about our paper.

My question and two comments are as follows:

1. In Figure 1a, there is a protein migrating near the 250 kDa marker, which is present only in Piezo1-GST incubated C2Cl2 cell extract, as SERCA2. Why did authors choose SERCA2 over this protein while searching for the interacting partners of Piezo1?

Response: We thank the reviewer for the comment. Actually, we have also excised the gel near the 250 kDa marker for mass spectrometry and identified peptides of Piezo1, indicating that this band might represent partially degraded Piezo1 proteins. We have included these information in the revised manuscript.

2. Figure legend 4d is missing

Response: We thank the reviewer for pointing this out. We have included the legend for the revised Fig. 6a. “a and g, Representative images showing the migrated HUVEC cells in the transwell assay.”

3. Authors use “mechanosensitive (MS) ion channels” at the beginning of the text. However, when explaining patch clamp results, they switch to the abbreviation “MA” (Mechanically-activated) current of these channels. Please use one of them throughout the text.

Response: We thank the reviewer for the suggestion. To be more specific about the way for inducing Piezo1 currents, we have used either poking-induced or stretch-induced Piezo1 currents in the revised paper.

REVIEWERS' COMMENTS:

Reviewer #1 (Remarks to the Author):

This is an exceptionally strong revision. The authors fulfilled all my requests for experiments, data removal, and manuscript writing. With their additional experiments using a synthetic peptide the authors exceeded my expectations.

I strongly recommend accepting what is now a beautiful manuscript.

Reviewer #2 (Remarks to the Author):

The authors have improved their manuscript considerably and I would in principle recommend publication of the revised MS.

There is one thing that should be corrected and if necessary also reanalyzed.

When the authors plot data from poking experiments they have plotted I_{max} values. These should all be corrected for cell capacitance (I_{max}/pf). this reanalysis should be done for all data using poking (all data in Figure 4 and Figure 5 for instance).

Poking data shown in supplementary data should be treated the same way.

Reviewer #3 (Remarks to the Author):

I am satisfied with the answers of the authors to my questions.

Reviewer #1 (Remarks to the Author):

This is an exceptionally strong revision. The authors fulfilled all my requests for experiments, data removal, and manuscript writing. With their additional experiments using a synthetic peptide the authors exceeded my expectations. I strongly recommend accepting what is now a beautiful manuscript.

Response: We thank the reviewer for the positive comments and recommendation for publication of our paper.

Reviewer #2 (Remarks to the Author):

The authors have improved their manuscript considerably and I would in principle recommend publication of the revised MS.

Response: We thank the reviewer for the positive comments and recommendation for publication of our paper.

There is one thing that should be corrected and if necessary also reanalyzed.

When the authors plot data from poking experiments they have plotted I_{max} values. These should all be corrected for cell capacitance (I_{max}/pF). this reanalysis should be done for all data using poking (all data in Figure 4 and Figure 5 for instance).

Poking data shown in supplementary data should be treated the same way.

Response: We thank the reviewer for the suggestion. We have re-analyzed the data and presented the poking-induced currents as pA/pF in Fig. 4, 5, Supplementary Fig. 2, 3, 5, 6.

Reviewer #3 (Remarks to the Author):

I am satisfied with the answers of the authors to my questions.

Response: We thank the reviewer for the positive comments and recommendation for publication of our paper.